# Two Sides of the Same Coin for Health: Adaptogenic Botanicals as Nutraceuticals for Nutrition and Pharmaceuticals in Medicine

**DOI:** 10.3390/ph18091346

**Published:** 2025-09-08

**Authors:** Alexander Panossian, Terrence Lemerond

**Affiliations:** 1Phytomed AB, Sjöstadsvägen 6, 59344 Västervik, Sweden; 2EuroPharma USA Inc., Green Bay, WI 54311, USA; terryl@europharmausa.com

**Keywords:** botanicals, dietary supplements, pharmaceuticals, adaptogens, network pharmacology, food and drug regulation

## Abstract

**Background:** Adaptogens, commonly used as traditional herbal medicinal products for the relief of symptoms of stress, such as fatigue and exhaustion, belong to a category of physiologically active compounds related to the physiological process of adaptability to stressors. They are used both as pharmaceuticals in medicine and as dietary supplements or nutraceuticals in nutrition, depending on the doses, indications to treat diseases, or support health functions. However, such a dual-faced nature of adaptogens can lead to inconsistencies and contradictory outcomes from Food and Drug regulatory authorities in various countries. **Aims:** This narrative literature review aimed to (i) specify five steps of pharmacological testing of adaptogens, (ii) identify the sources of inconsistencies in the assessment of evidence the safety, efficacy, and quality of multitarget adaptogenic botanicals, and (iii) propose potential solutions to address some food and drug regulatory issues, specifically adaptogenic botanicals used for prevention and treatment of complex etiology diseases including stress-induced, and aging-related disorders. **Overview:** This critically oriented narrative review is focused on (i) five steps of pharmacological testing of adaptogens are required in a sequential order, including appropriate in vivo and in vitro models in animals, in vitro model, and mechanisms of action by a proper biochemical assay and molecular biology technique in combination with network pharmacology analysis, and clinical trials in stress-induced and aging-related disorders; (ii) the differences between the requirements for the quality of pharmaceuticals and dietary supplements of botanical origin; (iii) progress, trends, pitfalls, and challenges in the adaptogens research; (iv) inadequate assignment of some plants to adaptogens, or insufficient scientific data in case of *Eurycoma longifolia*; (v) inconsistencies in botanical risk assessments in the case of *Withania somnifera*. **Conclusions:** This narrative review highlights the importance of harmonized standards, transparent methodologies, and a balanced, evidence-informed approach to ensure consumers receive effective and safe botanicals. Future perspectives and proposed solutions include (i) establish internationally harmonized guidelines for evaluating botanicals based on their intended use (e.g., pharmaceutical vs. dietary supplement), incorporating traditional use data alongside modern scientific methods; (ii) encourage peer review and transparency in national assessments by mandating public disclosure of methodologies, data sources, and expert affiliations; (iii) create a tiered evidence framework that allows differentiated standards of proof for traditional botanical supplements versus pharmaceutical candidates; (iv) promote international scientific dialogs among regulators, researchers, and industry to develop consensus positions and avoid unilateral bans that may lack scientific rigor; (v) formally recognize adaptogens a category of natural products for prevention stress induced brain fatigue, behavioral, and aging related disorders.

## 1. Introduction

Adaptogens belong to a category of physiologically active compounds or botanicals that trigger physiological processes of adaptability to stressors [1,2,3]. More than 110 medicinal plants were found to exhibit adaptogenic activity [2]. The number of plants reported as being adaptogenic has increased exponentially during the past decades. However, only a few of them comply with the most characteristic features, as shown in Table 1, an upgrade from [2].

In the first definition of adaptogens dated 1958, a toxicologist, Lasarev, focused on the so-called “state of non-specific resistance of organism” [4,5] that is associated with the notions of “adaptability” [6,7] defined by Georges Cunghilaim in 1943 and “stress”, Cannon [8] and Hans Selye [9,10]. In 1967, Brekhman and Dadimov implemented the term adaptogens in pharmacognosy, postulating that adaptogens are innocuous medicinal plants, normalizing body functions in response to various harmful environmental conditions [11,12], as shown in Table 1. The first adaptogens extensively studied were Ginseng, probably its reputation as a panacea for all diseases [13,14], Eleutherococcus (Siberian ginseng), and later *Rhodiola rosea* (Golden root) due to their tonic effects [15,16,17,18,19,20,21,22]. These postulates were supported by numerous publications during the last decades [1,2,3], resulting in updated definitions of adaptogens, see Table 1 and Table 2 [2,23], included in textbooks of pharmacognosy and phytotherapy [24,25,26,27,28,29,30,31,32], due to tremendous progress in biomedical sciences. However, new evidence is required to support the safety of some adaptogens, e.g., Withania, Bryonia, and some other putative adaptogens, such as Sideris, Eurocoma, Hypericum, etc., supporting Brekhmans’ postulations and current definitions of adaptogens.

**Table 1 pharmaceuticals-18-01346-t001:** Summary of characteristics of adaptogens *.

*Definition*: Adaptogens are natural compounds or plant extracts that enhance the adaptability and survival of living organisms in response to stress. *Therapeutic category and action*: adaptogens, tonic, immunomodulatory [4,15,16,17,18,19,20,21,22,24,25,26,27,28,29,30,31,32,33,34,35,36,37,38,39,40,41,42,43,44,45,46,47,48,49], *Chemical class*: predominantly tetracyclic triterpene, phenethyl- and phenylpropanoid glycosides, lignans, etc. *Pharmacological activity*: adaptogenic, tonic, stress-protective, stimulating *Mechanism of action*: multitarget effects on the neuroendocrine-immune system; adaptogens are adaptive stress response modifiers of cellular and organismal defense systems, activating intracellular and extracellular adaptive signaling pathways, expression of stress-activated proteins, neuropeptides, antioxidant enzymes, and anti-apoptotic proteins of an organism, resulting in nonspecific resistance to various stressors and increased survival [1,2,3,14,23,33,34,35,36].

*—Reproduced from Panossian et al. [2] with minor modifications.

Depending on indications for their use to treat diseases, or health-supporting functional claims, they can be classified both as drugs in medicine and as dietary supplements or nutraceuticals in nutrition, e.g., *Panax ginseng* C.A.Mey. roots [47,48], Schisandra chinensis (Turcz.) Baill., fruit [38], *Sedum roseum* (L.) Scop. (syn. *Rhodiola rosea* L.) [48]. In this context, adaptogens occupy a unique and complex position at the intersection of food and pharmaceutical regulation [50,51], different from other categories of physiological compounds like vitamins, minerals, or antioxidants, which are essential for various physiological processes in the human organism.

A concept underlying preventive treatment for “subhealth” conditions (“*fu zheng*” in TCM for strengthening body resistance and “vital energy”—*qi*) by adaptogens was implemented in the USSR under the names Medical Fitness, Pharmacomacia, and Valeology [2,50,51]. The dual-faced nature of some botanicals used as pharmaceuticals and dietary supplements for nutrition can lead to inconsistencies and contradictory outcomes from the Food and Drug regulatory authorities in various countries. For instance, reports from Danish and Indian food authorities on the safety of Ashwagandha [52,53,54,55] reveal inconsistencies and controversial conclusions. As a result, this can raise conflicts of interest between the pharmaceutical and food industry and the Food and Drug regulatory authorities [56].

This narrative overview is an extension to our previous review articles on adaptogens, aiming to identify the sources of inconsistencies in the assessment of evidence, ensure the safety, efficacy, and quality of botanicals, and propose potential solutions to address the issues, particularly in the field of multi-component Botanicals and Botanical Herbal Products dealing predominantly with complex etiology, e.g., fatigue, adaptation in stress-induced, and aging-related disorders and diseases, see Table 1. These reviews provided comprehensive information on adaptogens’ research [1,2,3,15,33,34,35,36,41,42,43,44,45,46,47,48,49], including results of clinical studies. In this critically oriented narrative literature review, we raised two simple questions:How to distinguish adaptogens from other herbs that are claimed as putative adaptogens, e.g., *Eurycoma longifolia* [57]?What are the sources of controversial conclusions from the assessment of efficacy and safety of the same plant in various countries, e.g., *Withania somnifera* in India and Denmark [52,53,54,55,56]?

We also hypothesized that this is due to insufficient evidence and differences in the criteria for assessing the quality, efficacy, and safety of botanicals, which may lead to inconsistent results, particularly in clinical studies.

## 2. Progress, Trends, Pitfalls, and Challenges in the Adaptogens Research

Adaptogens are botanicals that enhance adaptability, resilience, resistance, and survival in response to stress-induced and aging-related disorders [2,3].

The term adaptogen is derived from adaptation, which is coined for the physiological process of adaptability [6,7], Figure 1, of the organism to repeated action of the botanical that triggers adaptive stress response via intracellular and extracellular adaptive signaling pathways and networks in neuroendocrine-immune, cardiovascular, and gastrointestinal systems and resulting in increased adaptability, resilience, resistance, and survival in stress-induced and aging-related disorders [1,2,3].

Adaptogens play a similar role in defending the plant against environmental challenges, including viruses, harmful bacteria, insect-borne diseases, excessive UV rays, and the physiological ravages of chronic stress [3].

Initially, the postulated definition of adaptogens was modified several times (Table 2), as our understanding of their mode and mechanisms of action, and the progress in biomedical sciences over the last few decades, improved [1,2,3]. Witnessing the progress in stress research, the evolution of the adaptogen concept and milestones of adaptogen research [59,60] reflects scientific knowledge to new findings over time: from discoveries of hormones, neurotransmitters, molecular chaperones (heat shock proteins), neuropeptide NPY, eicosanoids, and other mediators of intracellular and extracellular communications in response to adaptive hormetic response, to network pharmacology of OMICs and systems biology concepts [1,2,3].

**Table 2 pharmaceuticals-18-01346-t002:** Definitions of adaptogens *.

• Adaptogens are medicinal substances causing the “state of nonspecifically increased resistance” of the organism. • Only those preparations that meet the following requirements may be included in the group of adaptogens: (a) An adaptogen should be innocuous and cause minimal disorders in the physiological functions of an organism; (b) The action of an adaptogen should be nonspecific, i.e., it should increase resistance to adverse influences of a wide range of factors of physical, chemical, and biological nature, (c) An adaptogen may possess normalizing action irrespective of the direction of the foregoing pathologic changes. • The adaptogens are nontoxic compounds with polyvalent mechanisms of action and pharmacological effects related to adaptability and survival. • Adaptogens are substances that elicit in an organism a state of nonspecifically raised resistance, allowing them to counteract stressor signals and to adapt to exceptional strain. • Adaptogens are metabolic regulators that increase the ability of an organism to adapt to environmental factors and to avoid damage from such factors. • Plant adaptogens are agents that reduce the damaging effects of various stressors due to the reduction in the reactivity of the host defense system. They adapt organisms to stress and have a curative impact on stress-induced disorders • Adaptogenic substances have the capacity to normalize body functions and strengthen systems compromised by stress. They have a protective effect on health against a wide variety of environmental assaults and emotional conditions. • Adaptogens comprise a pharmacotherapeutic group of herbal preparations used to: increase attention and endurance in fatigue and prevent/mitigate/reduce stress-induced impairments and disorders related to neuro-endocrine and immune systems. • Botanical adaptogens are plant extracts, or specific constituents of plant extracts, which function to increase survival in animals and humans by stimulating their adaptability to stress by inducing adaptive responses. • Adaptogens are stress-response modifiers that increase an organism’s nonspecific resistance to stress by increasing its ability to adapt and survive. • Botanical adaptogens are metabolic regulators that increase survival by increasing adaptability in stress. • Adaptogens are natural compounds or plant extracts that increase the adaptability and survival of living organisms to stress. • Adaptogen—any of various natural substances used in herbal medicine to normalize and regulate the systems of the body. https://www.dictionary.com/browse/adaptogen (assessed on 3 September 2025). • Adaptogens are currently defined as a therapeutic category/ pharmacological group of herbal medicines or/and nutritional products, increasing adaptability, survival, and resilience in stress and aging by triggering intracellular and extracellular adaptive signaling pathways of cellular and organismal defense systems (stress system, e.g., neuroendocrine-immune complex • Adaptogens act as chronic eustress, activating adaptive stress response, resilience, and overall survival. Adaptogens trigger the defense adaptive stress response of the organism to stressors, leading to the extension of the limits of resilience to overload, brain fatigue, and mental and aging disorders [3] • The term “adaptogen” refers to a time-dependent physiological adaptation process of the organism in response to the repeated administration of a plant substance that activates intracellular and extracellular adaptive signaling pathways and triggers an adaptive stress response [23].

*—Reproduced from Panossian et al. [2] with additions.

Overall, the mechanism of action of adaptogens remains largely unexplored, despite significant progress in brain and endocrine research, particularly in the regulation of biorhythms, which are closely linked to senescence and aging processes (Figure 2 and Figure 3) [1,2,3]. Several hypothetical models of time [1,2,3,12,61,62,63,64,65,66] and dose [3] dependent relationship effects of adaptogens were considered based on adaptive stress response and homeostasis concepts [1,2,3].

Considering the chemical structure of adaptogenic compounds, they were distinguished into three groups, suggesting that their pharmacological activity is associated with effects on the neuroendocrine immune complex and multitarget pleiotropic effects [1,2,3,15]. They include

(i)Steroids-like tetracyclic and pentacyclic triterpenes, such as ginsenosides, withanolides, cucurbitacines, and andrographolides, which structurally resemble the catabolic hormones corticosteroids that inactivate the stress system to protect against overreaction to stressors; and anabolic-androgenic steroids structurally related to testosterone and estrogen androgenic steroid hormones which regulate lipids and sugar metabolism via hypothalamus–pituitary–adrenal hypothalamus–pituitary–adrenal (HPA)-axis = hypothalamus–pituitary–adrenal axis [1,2,3,15,67,68].(ii)Catecholamine-like phenolic compounds such as phenylpropanoids, phenylethane derivatives, and lignans, having similar to neurotransmitter hormones pharmacophores and suggesting an effect on the sympathetic nervous system, possibly implying an impact in the early stages of the stress response [1,2,3],(iii)Oxylipins [2,15,69] (unsaturated C-18 trihydroxy and epoxy polyhydroxylated unsaturated fatty acids), homologous to anti-inflammatory resolvins and pro-inflammatory leukotrienes [70], which are involved in the pathogenesis of Alzheimer’s disease. *Rhodiola rosea*, *Withania somnifera*, and *Eleutherococcus senticosus* downregulate the expression of key genes (ALOX5AP, DPEP2, LTC4S) involved in the biosynthesis of leukotrienes A, B, C, D, and E, resulting in inhibition of the leukotriene signaling pathway, suggesting their potential benefits in Alzheimer’s disease [3,71].

Figure 3 illustrates the effects of adaptogens on key mediators of the neuroendocrine-immune complex, cardiovascular, and detoxification systems that regulate the adaptive stress response to stressors/pathogens in stress- and aging-induced diseases and disorders. CRH- and ACTH-induced stimulation of GPCR receptors activates the cAMP-dependent protein kinase (PKA) signaling pathway, regulating energy balance and metabolism across multiple systems, including adipose tissue (lipolysis), liver (gluconeogenesis and glucose tolerance), pancreas, and gut (insulin exocytosis and sensitivity). The key molecules involved in the PI3K-Akt signaling pathway are receptor tyrosine kinases. Activating the PI3K-Akt signaling pathway promotes cell proliferation and growth, stimulates cell cycle, vascular remodeling, and cell survival, and inhibits cell apoptosis in response to extracellular signals. The nonspecific antiviral action of adaptogens is associated with the activation of innate immunity by upregulation of the expression of the pathogen’s pattern recognition receptors, specifically toll-like receptors and TLR-mediated signaling pathways. The protein kinase C (PKC) family of enzymes, with its various isoforms, plays a cell-type-specific and essential role, particularly in the immune system, through NF-κB activation [3].

Three stress-activated MAPK signaling pathways, which play crucial roles in cell proliferation, differentiation, survival, and death, have been implicated in the pathogenesis of numerous human diseases, including Alzheimer’s disease, Parkinson’s disease, and cancer.

The stress factors inducing the activation of the c-Jun N-terminal kinase (JNK)/stress-activated protein kinase (SAPK)-mediated adaptive signaling pathway are heat shock, irradiation, reactive oxygen species, cytotoxic drugs, inflammatory cytokines, hormones, growth factors, and other stresses. Activating the JNK/MAPK10 signaling pathway promotes cell death and apoptosis by upregulating proapoptotic genes.The activation of the extracellular-signal-regulated kinase (ERK) pathway is initiated by hormones and stresses to trigger endothelial cell proliferation during angiogenesis, T cell activation, long-term potentiation in hippocampal neurons, phosphorylation of the transcription factor p53, activation of phospholipase A2 in mast cells, followed by activation of biosynthesis of leukotrienes and inflammation/allergy, etc.The third major stress-activated p38 signaling pathway contributes to the control of inflammation, the release of cytokines by macrophages and neutrophils, apoptosis, cell differentiation, and cell cycle regulation [3].

### 2.1. Stress-Protective, Stimulating, and Tonic Activity of Adaptogens

The stress-protective effect is a result of feedback downregulation mechanisms following single-dose stimulation of an adaptogen (a mild pro-stressor) and repeated administration of plant adaptogens or physical exercise (tonic effect). One of the adaptive stress-response mechanisms of feedback regulation in organisms is the HPA-mediated loop of feedback down-regulation in anti-inflammatory response, as shown in Figure 2, where the key regulator is cortisol [61,63,65,68]. Adaptogens trigger various alternative signaling pathways via stress-adaptive feedback networks (see Figure 3, refs. [1,2,3]).

Stress as a defense response of an organism to external factors (strain) stimulates the formation of endogenous activating messengers, such as catecholamines, prostaglandins (PG), cytokines, nitric oxide (NO), and PAF (“switch on”-system), which in turn activate energy and metabolic resources of the organism. For example, NO is produced in large quantities during host defense and immunologic reactions. Corticosteroids, CRF, Heat shock proteins (Hsp), and PGE2 are endogenous mediators of cellular communications, which protect cells and whole organisms from overreacting to the activating messengers (“switch off” system) [61,68].

The balance between the activities of the “switch on” and “switch off” systems—“reactivity”—reflects the organism’s sensitivity to stressors and its level of protection against their damaging effects, “homeostasis”. It appears that, in the process of adapting to the impact of stressors, this balance is shifted to a higher level of equilibrium and homeostasis. Adaptogens increase the production of both activating and deactivating messengers of the stress system. In other words, adaptogens increase the capacity of the stress system to respond to external signals at the higher level of equilibrium-heterostasis [61,68].

Thus, a single administration of an adaptogen primarily produces a challenging (stimulating or stress-agonizing) effect, a fact that is utilized in sports medicine, where a single dose of an adaptogen can enhance athletes’ performance by making them more alert [62,63,64,65,72]. The stimulating effects of a single dose of adaptogens are primarily due to interactions within the central nervous system (CNS) and hypothalamic–pituitary–adrenal (HPA) axis but do not require prolonged processes such as gene and protein expression, neurogenesis, cell proliferation, and differentiation [1,2,3,62,63,64,65]. The tonic effect of adaptogens is due to their repeated administration at multiple doses [3,15,63].

Plant adaptogens stimulate the nervous system through mechanisms that are quite different from those of conventional stimulants. Depending on the mediators of the stress system involved in the adaptogen-induced stress response, an immediate (single-dose effect) or a long-term (following multiple administrations) stimulating effect may be observed [65,73].

There are essential differences between adaptogens and conventional stimulants of the CNS, such as caffeine, nicotine, amphetamines, and cocaine, which produce a sense of euphoria and can be used to increase alertness and the ability to concentrate on mental tasks, boosting endurance and productivity [2,24,65]. However, long-term stimulant abuse can impair cognitive function and lead to psychotic symptoms and drug dependence. In contrast, adaptogens which exhibit high rate of recovery process after exhaustive physical load, low energy depletion, increased performance in stress and survival in stress, good quality of arousal, lack of senescence, rare side effects, upregulation of expression DNA/RNA and proteins, increased Hsp70 synthesis, and activation of NPY production, which activate HPA -axis [2,65,74,75].

Adaptogens exhibit stress-protective and tonic effects [15,16,17,18,62,65], as shown in Figure 4a. After repeated dose administration, they trigger adaptive stress response signaling pathways and mediate gene expression in the HPA axis and other body tissues [1,2,3,61,65,74,75], as shown in Figure 2 and Figure 3. The stress-protective effect achieved by multiple administrations of adaptogens is not the result of inhibiting the stress response of an organism but rather of adaptive changes in the organism in response to the repeated stress-agonistic effect of the adaptogens [1,2,3,62,68,76,77,78]. Importantly, adaptogens exhibit anti-stress effects by activating cellular adaptive mechanisms, acting as eustressors or challengers (i.e., “good stressors”), mild stress mimetics, agonists, or “stress vaccines” that induce stress-protective responses, but not stress antagonists [1,2,3,24,68,76,77,78].

Adaptogens-induced outcomes and changes in various endpoints from the baseline over time (Figure 4a) depend on the baseline: (i) a state of rest, and (ii) under various stressors. Under rest, they exhibit a stimulating impact on a single dose and a tonic effect with repeated multiple administration (preconditioning), similar to physical exercise (preconditioning). Against the background of stress (physical exertion or mental stress, preconditioning), adaptogens exhibit a stress-protective effect by reducing the threshold of sensitivity to the stressor and thereby extending the duration of the protective effect in time until the fatigue phase, until complete exhaustion or damaging effect (distress) of continuous stressful exposure, e.g., in stroke [3].

The dose-effect relationship of adaptogens is typically characterized by a bell-shaped curve in many pharmacological models, and in some models, a biphasic curve emerges at excessively high doses [3] (Figure 4b). This aligns somewhat with the hormesis concept, characterized by a biphasic reversal effect, where positive (stimulating) effects at low doses transition to negative (toxic/harmful) effects at high doses. The drug-response interactions model has several other limitations regarding biphasic dose–response. The primary mechanism common to both conditioning a hormetic and adaptive response is that it activates molecular signaling pathways, enhancing the cell and organism’s ability to tolerate more severe stress. The optimal range of doses for adaptogens or conditioning treatments corresponds to the hormetic zone dose–response pattern. The underlying molecular mechanisms of hormesis are not fully understood. The theoretical background of hormesis is related to the hypothesis of interactions between biologically active compounds (ligands, interventions, or drugs) and two target proteins (receptors) that exhibit functionally opposite responses at different concentrations. It was proposed that a drug acting as a competitive antagonist at either or both of the receptors changes the relationship between the two opposing concentration-effect curves, resulting in potentiation, antagonism, or reversal of the observed effect; the theoretical model suggested that the total impact on the system can be obtained by the algebraic summation of the two effects resulting from the activation of the two opposing receptor populations, providing a classical hormesis biphasic curve. However, in practice, dose–response patterns are significantly complicated due to many other interactions, including: (i)—multiple targets (receptors) of different affinity to the active compound, (ii)—other regulatory proteins or mediators in the networks involved in the adaptive stress response, (iii)—feedback downregulations in the molecular signaling pathways and/or; (iv)—the metabolic transformation of active ligands into metabolites, a secondary ligand, has different affinities to various receptors of the adaptive stress response [35]. These observations demonstrate that a specific plant extract, characterized by a product-specific quantitative and qualitative composition, as well as an HPLC fingerprint/pharmaceutical profile, can exhibit varying pharmacological activity, profiles, and signatures depending on the dose, resulting in a dual response in the organism, ranging from positive to inactive or even negative [3,35].

### 2.2. What Is Necessary and Sufficient to Be Classified as an Adaptogenic Plant?

Five steps of pharmacological testing of adaptogens are required in a sequential order:An appropriate in vivo model in animalsAn appropriate in vitro modelMechanism of action by a proper biochemical assayMechanism of action by an appropriate molecular biology technique in combination with network pharmacology analysisClinical trials in stress-induced and aging-related disorders

Pharmacologic assessment of adaptogenic activity is commonly used in various animal stress tests involving exposure to cold, heat, altered atmospheric pressure and oxygen content, radiation, toxic substances, starvation, fear, and chronic diseases. It has been demonstrated that the primary feature of adaptogens is their ability to enhance resistance to both physical and emotional stress [15]. The most suitable pharmacological methods for assessing adaptogenic activity are chronic unpredictable stress (CUS) animal models, which induce affective behaviors in mice and, once established, measure stress-related alterations in the intrinsic excitability and synaptic regulation of the medial prefrontal cortex layer 5/6 pyramidal neurons. Adult male mice received 2 weeks of ‘less intense’ stress or 2 or 4 weeks of ‘more intense’ CUS, followed by assessment of sucrose preference for anhedonia and the elevated plus maze for anxiety. They forced a swim test for evaluation of depressive-like behaviors. An intense CUS exposure results in increased anhedonia, anxiety, and depressive behaviors, while less intense stress results in no measured behavioral phenotypes [80].

Adaptogens activate multiple cytoprotective mechanisms that increase cell survival (antioxidant, immune modulation, Hsp70 modulation [1,2,3]), and trigger the generation of hormones (e.g., corticotropin, cortisol, gonadotropin-releasing hormones, urocortin, melatonin [33]), neuropeptide N [75], and neurohormones, and neurotransmitters playing a key role in metabolic regulation and homeostasis [3], Figure 2.

Adaptogens trigger an adaptive stress response, activating adaptive signaling pathways, e.g., G-protein coupled (GPCR), tyrosine, toll-like receptors, and I3PK-mediated pathways that are known to promote survival in response to stress, suggesting neuroprotective activity and potential benefits of adaptogens in neurodegenerative diseases, Figure 4 [1,2,3,33].

Overall, adaptogens are active in numerous conditions and diseases associated with stress and aging-related impairments of the neuroendocrine-immune complex, as well as energy and fatigue, which must be tested clinically in human subjects [2,3].

### 2.3. Progress and Trends in Adaptogens Research

Breakthroughs in our knowledge of the mechanisms of action of adaptogens are usually associated with the discovery of previously unknown mediators of intracellular and intercellular communications and their physiological functions and mechanisms of action, such as molecular chaperones of neuropeptide Y and many new proteins involved in the regulation of homeostasis and the development of various diseases and their symptoms. For instance, in post-stroke and traumatic long-lasting brain or mental fatigue associated with impaired angiogenesis and neurogenesis during recovery [3]. An invaluable breakthrough was the application of network pharmacology methods, leveraging the latest techniques in molecular biology, metabolomics, proteomics, transcriptomics, genomics, and microbiomics to study interactions.

Adaptogens are characterized as botanicals with polyvalent action and pleiotropic activity, often referred to as a “Ginseng-like panacea for all diseases [13,14].” Progress in biomedical science, particularly in network pharmacology, metabolomics, proteomics, transcriptomics, genomics, and microbiomics—collectively known as omics—and systems biology, has significantly contributed to uncovering the mechanisms of action of adaptogens and enhancing our understanding of their effects [3,14,23,33,34,35,36,81,82,83,84,85,86,87].

Network pharmacology is an emerging discipline that integrates systems biology, bioinformatics, and pharmacology to understand the actions of drugs on a systemic, network level. Instead of the traditional “one drug–one target” model, network pharmacology employs the “multitarget, multi-component” paradigm, which is particularly useful for complex diseases and multi-herbal medicines, Table 3 [3,14,33,35,81].

Network pharmacology offers a robust systems-level framework for understanding drug mechanisms, especially in complex, multitarget conditions. While it introduces a more realistic model of pharmacological action, it must overcome limitations like data incompleteness, lack of dynamism, and validation gaps. For mechanistic insights, it should ideally be integrated with experimental biology and temporal models to achieve both predictive power and clinical relevance [81].

The advancement of integrative omics network pharmacology and artificial intelligence in natural products has opened new avenues for the following areas:elucidation of the mechanisms of action of medicinal plants [3,14,33,34,35,36,71,81,82,83,85,88,89,90,91];understanding the synergistic therapeutic actions of complex bioactive components in medicinal plants [3,33,34,35,92,93];providing a rationale for traditional Chinese medicine, as well as enhancing the quality of TCM drug research and the speed and efficiency of developing new TCM products [94,95,96,97,98,99];discovering and developing new botanical hybrid combinations [3,14,35];predict drug–herb interactions [100], adverse events [101], and potential toxic effects [102,103,104,105].

Unlike conventional medicines, botanical adaptogens comprise multi-component active compounds and phytochemicals, whose interactions lead to novel and unexpected pharmacological activities due to their synergistic and antagonistic effects [14,33,35,81,106]. Emerging research also highlights how extraction methods can significantly influence the physicochemical and functional characteristics of bioactive plant-derived components, such as dietary fibers from *Rubus chingii* Hu., suggesting that extraction processes may similarly impact the efficacy and bioavailability of botanicals [107].

Similarly, the pharmacological activity of a combination of several plants or herbal extracts, such as those in botanical hybrid preparations (e.g., TCM formulas), results in a new biologically active substance with unique pharmacological characteristics [35].

Primary outcomes from gene expression studies of adaptogens in isolated brain cells indicate that adaptogens activate adaptive stress-response canonical pathways, modulating numerous physiological processes and the progression of stress-induced and aging disorders [3]. In addition, various adaptogens regulate the expression of many other genes involved in neurogenesis, angiogenesis, innate immunity, and inflammaging, which play a crucial role in stress-induced and aging-related disorders [2,93].

Network pharmacology provides a possible mechanism of action. Network analysis must be conducted in combination with bioinformatic databases and experimental pharmacology, based on the results of gene expression and transcriptomics arrays obtained in vitro or in vivo studies, and validated, e.g., by PCT technique.

Molecular docking approaches in silico studies are not a stand-alone technique. Without functional assays (e.g., enzymatic inhibition) and binding studies (e.g., surface plasmon resonance, isothermal calorimetry, or microscale thermophoresis), there is no evidence to support the accuracy of in silico results. Otherwise, they remain hypothetical.

Various challenges in network pharmacology for natural products and phytotherapy research, which should be addressed [81,106].

### 2.4. Pitfalls in Adaptogen Research: Inadequate Assignment of Some Plants to Adaptogens or Insufficient Scientific Data

Polyvalence of some medicinal plants often leads to unsupported claims that they are adaptogens, as is the case with *Eurycoma longifolia*. While it is recognized in traditional medicine across Southeast Asia and regulated in markets like the EU, the official pharmacopeial status appears to be limited to three countries: Malaysia, Indonesia, and Vietnam. Several publications claim that *Eurycoma longifolia* Jack is an adaptogen [108,109,110,111,112,113,114]. However, there is no scientific evidence supporting these statements. The authors refer to:Traditional use of “Malaysian Ginseng” as an aphrodisiac that stimulates sexual desire, antimalarial, anti-diabetic, antimicrobial, and antipyretic activities, and its polyvalent activity in various diseases [115,116],Gonadotropin-induced increase in testosterone in vitro studies and elevated blood testosterone in animals and human studies [57,110,117,118], possibly associated with an increase in male fertility and ergogenic activity [109,110,112,113,119,120,121].

However, that is not a sufficient characteristic criterion to classify Eurycoma as an adaptogen. See the definition and characteristics of adaptogens in the Amendment below.

Furthermore, the quality of clinical trials in humans, preclinical studies in animals, and in vitro studies is not satisfactory, making it difficult to draw robust conclusions on the efficacy and safety of Eurocoma products.

E.g., Talbott et al., 2013 [111] state that:Active compounds of Eurycoma are 4300 dalton glycopeptides consisting of 36 amino acids.

“*Eurycoma contains a group of small peptides referred to as “eurypeptides” with 4300 dalton glycopeptides 36 amino acids that are known to have effects in improving energy status and sex drive in studies of rodents. Animal studies have shown that many of the effects of the extract are mediated by its glycoprotein components. Typical dosage recommendations, based on traditional use and on the available scientific evidence in humans, including dieters and athletes, call for 50–200 mg/day of a water-extracted tongkat ali root standardized to 22% eurypeptides*.”

However, the publication does not contain any information about the analytical methods used for quantification, accuracy, precision, selectivity, or purity of the analytical markers, which are essential for ensuring reproducible quality and efficacy. There is nothing in that study [111] about numerous biological active plant secondary metabolites isolated from *E. longifolia* various extracts, e.g., quassinoids eurycomanone, 13α(21)-epoxyeurycomanone, 13,21-dihydroeurycomanone, 14,15 β -dihydroxy-klaineanone, longilactone and eurycomalacton, 9-methoxycanthin-6-one; Laurycolactone; Eurycolactone B; Eurycomalide A; Eurylactone; Longilactone; Eurycomalactone; Eurycomanone; Eurycomanol; Pasakbumin B; Hydroxyklaineanone; Biphenyl-neolignan; Quassin etc., which are known as poorly water-soluble compounds [115,116,120,121,122,123].

According to the authors Talbott et al., 2013 [111], the mechanism of action is related to “*the bioactive complex 4300 dalton glycopeptides (“eurypeptides” with 36 amino acids) has been shown to activate the CYP17 enzyme (17 alphahydroxylase and 17,20 lyase) to enhance the metabolism of pregnenolone and progesterone to yield more DHEA (dehydroepiandrosterone) and androstenedione, respectively [114]. This glycoprotein water-soluble extract of Eurycoma longifolia has been shown to deliver anti-aging and antistress benefits subsequent to its testosterone balancing effects*”.

The effects of tongkat ali in restoring normal testosterone levels appear to be less due to actually “stimulating” testosterone synthesis, but rather by increasing the release rate of “free” testosterone from its binding hormone, sex-hormone-binding-globulin (SHBG).

In this way, eurycoma may be considered not so much a testosterone “booster” (such as an anabolic steroid), but rather a “maintainer” of normal testosterone levels and a “restorer” of normal testosterone levels (from “low” back “up” to normal ranges). This would make eurycoma particularly beneficial for individuals with sub-normal testosterone levels, including those who are dieting for weight loss, middle-aged individuals suffering with fatigue or depression, and intensely training athletes who may be at risk for overtraining.”

That is not related to the known mechanisms of action of adaptogens on the neuroendocrine-immune complex; see Amendment below.

Furthermore, the results of a clinical study of *Eurycoma longifolia* (Physta^®^) water extract plus multivitamins show a lack of activity when *Eurycoma longifolia* (Physta^®^) was compared with placebo [109]: Results: “*there were no significant between-group differences, within-group improvements were observed in the SF-12 QoL, POMS and MMSQ domains”, which is associated with CNS activity of Eurycoma longifolia (Physta^®^) water extract plus multivitamins on quality of life, mood and stress*”.

This clearly shows a lack of effect of *Eurycoma longifolia* (Physta^®^) on the CNS, suggesting that Eurycoma is not an adaptogen.

There are many shortcomings in conducted randomized, placebo-controlled, double-blind clinical trials [109,110,112,119,120,121,124], which are not in line with CONSORT recommendations [125,126], including insufficient information regarding:Insufficient description of the study medication (see above),Randomization (a method used to generate the random allocation sequence, including details of restriction)Implementation (who generated the allocation sequence, enrolled participants, assigned participants to their groups, etc.)Blinding (preparation had the same appearance, test, and odor as placebo; how care providers, those assessing outcomes, were blinded; how the success of blinding was evaluated)Allocation concealment (the mechanism used to implement the random allocation sequence, such as sequentially numbered containers, describing any steps taken to conceal the sequence until interventions were assigned; it is not clear whether the sequence was concealed until interventions were assigned),Procedure for treatment compliance (how measurements of compliance of individual patients with the treatment regimen under study were documented).Monitoring,Settings and locations where the data were collected,Quality assurance and quality control,Deviations from the protocolSelective reportingThe trial was conducted per ICH guidelines for GCP.Voucher specimen (i.e., retention sample was retained and, if so, where it is kept or deposited).The role of the study sponsor/funder,Inappropriate statistical tools and statistical analysis (e.g., lack of between-groups comparison of changes from the baseline by two-way ANOVA, etc.).

Other limitations of the studies, e.g., Muniandy et al., 2023 [124]:

⇒ a purposive sampling in a randomized, double-blinded, placebo-controlled, parallel-group study.

⇒ lost to follow-up and missing data points

⇒ the lack of nutritional intake information, which can be a limitation for a comprehensive analysis of the potential influence of dietary factors on the observed outcomes.

Overall, there is no convincing evidence yet to support the claim that Eurocoma is an adaptogen.

### 2.5. Dual-Use Dilemma and Inconsistencies in Botanical Risk Assessments in the Case of Withania somnifera

Below is an insight into “*the two sides of the same coin*” and double standards in the assessment of the safety of *Withania somnifera* (L.) Dunal (Ashwagandha/Winter cherry root), resulting in controversial conclusions in India and Denmark [52,53,54,56,127,128].

Ashwagandha is the most popular plant in India and has been used in the traditional medical system (Ayurveda) to treat many diseases and health conditions for thousands of years. Due to its polyvalent pharmacological and adaptogenic activity, it is also known as Indian Ginseng [129,130,131,132]. Numerous preclinical and clinical studies have been conducted with several Ashwagandha preparations to provide evidence on their efficacy and safety in various conditions [133,134,135,136,137,138,139,140,141,142,143,144,145,146,147,148,149,150,151,152,153]. However, in most clinical studies, the authors typically conclude that “*The findings from the included studies indicate that Ashwagandha formulations have beneficial effects on stress and anxiety. The adverse effects associated with Ashwagandha are limited; however, further information is required to determine its safety with long-term administration*” [139].

*Withania somnifera* (L.) Dunal. is recognized in several national pharmacopeias and regulatory frameworks worldwide, though its status varies by country, including:In India, *Withania somnifera* is officially included in the Indian Herbal Pharmacopeia [154] where the monograph outlines: (i)—the plant names, (ii)—geographical distribution, (iii)—macroscopic and microscopic description of the roots, (iv)—chemical constituents (steroidal lactones including withanone, withaferin A, withanolides I, II, III, A, D, E, F, G, H, I, J, K, L, M, WS-L, P, and S, withasomnidienone, withanolide C, and alkaloids viz., cuscohygrine, anahygrine, tropine, pseudotropine, anaferine, isopellatierine, 3-tropyltigloate), (v)—Assays/analytical methods including HPLC conditions fingerprints, identifying withaferin A in extracts and withanolide J in vitro culture, (v)—quantitative standards (including total alkaloids (in total about0,2%), (vi)—adulteration, (vii)—pharmacology section, (viii)—reported activities including antistress, immunomodulatory, anticancer, antioxidant, an-ticonvulsive, anthelminthic, antiarthritic, chemopreventive, antibacterial, cardiopro-tective, antidepressant, antitoxic, hypoglycemic, diuretic, hypercholesterolemic, im-munosuppressive, antiradical, and adaptogenic activities, and (ix)—therapeutic category: adaptogen.In Europe, EMEA conducted a comprehensive literature search and reviewed available data, including information on the market in the European Union obtained from HMPC members. This review concluded that the requirements for establishing a Community herbal monograph on traditional herbal medicinal products containing *Withania somnifera* (L.) are met. Dunal, radix are not fulfilled since:
○the requirement of the definition of the ‘herbal preparation’ as the HMPC has not been able to find adequate evidence allowing a description of the herbal preparations (insufficient extract specification according to pharmaceutical requirements,○lack of adequate evidence allowing a demonstration of at least 30 years of medicinal use, including at least 15 years in the European Union.

The public statement dated 9 July 2013 claims that HMPC claims that a Community herbal monograph on *Withania somnifera* (L.) Dunal, radix cannot be established at present [155,156].

Denmark has banned Ashwagandha, citing safety concerns. This decision has been criticized by various scientific communities advocating for evidence-based evaluations [52,53].In Sweden, the regulation of Ashwagandha is decentralized, allowing local authorities to make decisions regarding its use. This approach permits its availability under certain conditions [157,158].In the U.S., Ashwagandha is permitted as a dietary supplement. The United States Pharmacopeia (USP) provides guidelines for its quality control, including High-Performance Liquid Chromatography (HPLC) methods to assess total withanolide content USP [159,160].Ashwagandha is allowed as a food supplement in the UK. The Medicines and Healthcare Products Regulatory Agency (MHRA) has approved clinical trials involving Ashwagandha, indicating its acceptance within specific regulatory frameworks [161].Ashwagandha is available as a dietary supplement in Germany. However, the Federal Institute for Risk Assessment (BfR) has recommended its inclusion in the EU’s list of substances for which safety has not been conclusively established, suggesting caution in its use [162].Ashwagandha is included in the Australian Register of Therapeutic Goods (ARTG), with over 320 listed medicines containing it, reflecting its acceptance in therapeutic products [163]Within the Association of Southeast Asian Nations (ASEAN), Ashwagandha is not uniformly included in national pharmacopeias. However, efforts are underway to harmonize traditional medicine regulations across member states. The ASEAN Common Technical Document (ACTD) framework is being utilized to standardize quality, safety, and efficacy requirements for herbal products, such as Ashwagandha [55].

A recent case in point is the Danish Technical University (DTU) Food Institute’s risk assessments of *Withania somnifera* (Ashwagandha) root [52,53], in which DTU identified possible thyroid and reproductive effects at high doses. These reports, which contributed to national-level restrictions and influenced regulatory positions in other countries, have been widely criticized by scientific and regulatory communities for methodological shortcomings and a lack of transparency [56,127]. The Danish agency cited concerns about hormonal effects and lack of a well-defined safe dose, and then concluded that this focus was controversial. As noted by the Danish Veterinary and Food Administration, its report argued no safe dose could be proven with existing data, a conclusion some experts find unconvincing [56,164]. Meanwhile, neither the EFSA [165] nor the FDA has taken similar actions; Ashwagandha remains a permitted supplement in both regions [164].

The views, scientific opinions, and conclusions drawn by Food and Drug authorities [52,53,54,165,166], dietary supplement manufacturers [167,168], experts in the field [169,170], distributors [171], independent reviewers [164], and researchers [3,35] differ and are somewhat contradictory.

#### 2.5.1. Food and Drug Authorities


In April 2023, Denmark banned Withania, stating that it is impossible to find a safe dose given the current data. The ban was based on a finding in 2020 by the Danish Technical University (DTU) that ashwagandha may induce abortions and has a possibly harmful effect on thyroid and sex hormones [52]. This decision was based on only one study in rodents, which found that ashwagandha reduced sperm quality and quantity [172], and one trial in humans suggested that ashwagandha might increase thyroid hormone levels [173], which could potentially cause delirium, heart failure, and dehydration [173].In Report of the Expert Committee constituted by the Ministry of Ayush, Govt. of India. The Central Council for Research in Ayurvedic Sciences, the experts state that “*Numerous safety studies consistently demonstrate that standardized Ashwagandha (Withania somnifera) root extract is safe for human consumption. The scientific data reveal that Ashwagandha root is well-tolerated across a wide range of doses, with no adverse outcomes reported in diverse demographic and clinical cohorts. Thus, the Expert Committee holds a strong view that the current evidence supporting the safety and efficacy of Ashwagandha root is robust, allowing it to be safely recommended and integrated into global health practices. This reinforces its standing as a safe herb in both traditional and contemporary health contexts; nevertheless, using correctly identified species is crucial, and stringent quality control should be carried out suitably in accordance with applicable regulatory norms. In view of its scientifically proven safety, the globally available and popular Indian herb Ashwagandha (Withania Somnifera) can be judiciously prescribed by the Ayurvedic physicians for its health benefits in humans* [54].
*The report by the DTU Food Institute (published in May 2020) needs a reconsideration due to two main reasons:*
○
*The report is not taking into consideration the different properties concerning function and safety between different plant parts of Ashwagandha*
○*There is evolving evidence on the safety and efficacy of Ashwagandha published since 2020 (more than 400 papers), DTU report* [52], *there are glaring instances of a lack of rigorous scientific scrutiny in the DTU risk assessment report, including conclusions drawn from publications of predatory journals”* [54].
The National Institute for Public Health and the Environment (RIVM) studied whether herbal preparations containing *Withania somnifera* are harmful to health, concluding that the herb can induce harmful effects in individuals who are sensitive to it. It is unknown which individuals are sensitive to *Withania somnifera*. As a precaution, RIVM advises consumers not to use herbal preparations containing *Withania somnifera*, especially during pregnancy [165].In contrast, the EFSA assessment report and scientific opinion support the safety of ashwagandha [166]. As quoted, “*In addition to its historical uses and pre-clinical studies, in the multiple published human clinical studies, ashwagandha and its preparations have been investigated in healthy adults, children, the elderly, and diseased populations. Although the primary objective of the majority of these trials was to investigate efficacy for health benefits, these studies provide insight into the potential safety and ‘tolerability’ of ashwagandha and its preparations in a diverse population. In healthy adults and children, Ashwagandha root has been used at doses of up to 5000 mg/day without adverse effects. Of the over 60 human clinical studies identified, more than 30 were placebo-controlled, double-blind clinical trials. As double-blind placebo-controlled trials are least likely to result in bias, these trials provide an opportunity to assess safety. Adverse effects of ashwagandha root or its preparations are rarely reported in humans. In some clinical trials, adverse effects were reported; however, the incidence was similar to that of the placebo group. A majority of the adverse events were transient and minor, and the frequency of these events was not related to the duration of intake. The findings from all of these studies support the safety-in-use of Ashwagandha root preparations* [166].


#### 2.5.2. Dietary Supplement Manufacturers

All Ashwagandha are not the same. All studies are product-specific, meaning they are conducted exclusively on the extract contained in KSM66. In all studies, the dose of 2 × 300 mg has been used, i.e., one capsule in the morning and one in the evening. Experience, combined with safety studies, shows that in cases of increased need, the dose can be doubled [168].In response to the DTU report, an expert report of the World Ashwagandha Council and Ixoreal Biomed Inc., Los Angeles, California, USA, and Heyderabad, India, presented data from 79 human clinical studies, 29 Toxicity studies, and 130 preclinical studies. The report also discusses observations by DTU Food Institute and presents data-driven responses to the comments. The results suggest that Ashwagandha root extract) It is clinically beneficial for reducing stress and anxiety by balancing cortisol levels, enhancing memory and cognition, increasing sex drive, endurance, and strength, as well as promoting muscle growth and recovery, and improving sleep quality and overall quality of life. The data assure its safety, as no serious adverse events have been reported in any of the clinical trials. Animal studies include acute and long-term toxicity assessment, including reproductive and developmental toxicity. The authors claim that KSM-66 Ashwagandha extract is safe and conclude that the report by the DTU Food Institute needs a reconsideration due to two main reasons:
○The report does not take into consideration the different properties concerning function and safety between different plant parts of Ashwagandha○There is evolving evidence on the safety and efficacy of Ashwagandha published since 2020 [167].

#### 2.5.3. Independent Reviewers

While these side effects are frightening, the report does not present a clear case for ashwagandha’s impact on hormones and fertility. The cases mentioned by the authors are rare, and many studies have found no evidence of thyroid problems. As far as abortion is concerned, the report presents no clinical evidence of ashwagandha as an abortifacient. Reference is made to the World Health Association’s advice that people should not consume ashwagandha during pregnancy or breastfeeding because “there is information that the root has been used as an abortifacient.” The origin of this information is unclear, but it was likely passed down as part of the Ayurvedic medical system. On the other hand, some journal articles state that ashwagandha is safe to use during pregnancy and was even used to boost health before and after birth. Without clear evidence, Demark took the most cautious option and banned ashwagandha altogether [164]. Whether or not consumers should follow Denmark’s ban is unclear. The concerns brought up by the DTU’s report are significant but not backed by substantial clinical evidence—especially the claims about ashwagandha inducing abortions. The European Medicines Agency and the FDA, two of the largest pharmaceutical regulating agencies, have yet to follow in Denmark’s footsteps by restricting or prohibiting ashwagandha. While this does not necessarily mean that Denmark is wrong about the potential side effects of ashwagandha, it does mean that the clinical data is limited enough that big pharmaceutical government agencies are not considering it [164].

Such a contradictory response to the regulatory requirements of the same plant products and their use in medicine is not surprising due to many objective and subjective reasons, including:○Dose-dependent reversal effect of Withania demonstrated in gene-expression study on cultivated brain cell culture, suggesting inhibition of corticotropin-releasing hormone CRH signaling pathways and inflammation in the concentration of 1.5 mg/L, and opposite effect—activation of CRH signaling pathway leading to the pro-inflammatory impacts in higher concentration of 5 mg/L [3,35]. A similar dose-dependent reversal effect of Withania was observed on the Electrophysiological Activity in Hippocampal Long-Term Potentiation, a Synaptic Model of Memory in the hippocampus slice model in vitro. In concentrations of 0.5–1.0 mg/L, Withania extract significantly increases pyramidal cell activity; however, at a higher dose of 2.5–10 mg/L, the effect decreases (Appendix A).○Inconsistencies result from studies dealing with diverse Withania extracts of diverse chemical composition, depending on numerous factors during harvesting, cultivation, methods of extraction of herbal substance, processing of herbal product, etc., ensuring sustainable quality and reproducible effects in pharmacological and clinical studies can lead to contradictory results.○Surprisingly, the quality of the studies of Withania, which raised concerns about the safety, was not scrupulously assessed in the DPU report. For example, the reporting of the clinical trial (registered at ClinicalTrials.gov, Identifier: NCT00761761) of Ashwagandha on thyroid and sex hormones [173,174,175] does not meet the CONSORT and ICH GCP requirements and cannot be taken into account for warning and banning Ashwagandha in Denmark and the Netherlands. The placebo/verum blinding and assignment procedures were not sufficiently described. Assessment criteria must be consistent across all studies.

Similar conclusions were drawn in a comprehensive review of preclinical and clinical studies examining the neuropsychiatric effects of WS, specifically its application in stress, anxiety, depression, and insomnia [176]. While benefits were seen in the reviewed studies, significant variability in the Withania root extracts examined prevents a consensus on the optimum Withania preparation or dosage for treating neuropsychiatric conditions. Withania generally appears safe for human use; however, it will be essential to investigate potential herb-drug interactions involving Withania if used alongside pharmaceutical interventions. Further elucidation of the active compounds of WS is also needed [176].

In summary, the incorporation of Ashwagandha into national pharmacopeias and its regulatory acceptance vary globally. While countries like India, the United States, and Australia have established standards or included it in official registers, others, such as France and Denmark, approach it with caution or restrictions. Ongoing efforts, particularly within ASEAN, aim to harmonize regulations for traditional herbal medicines, which could potentially influence Ashwagandha’s status in the future [55].

### 2.6. Key Issues Identified

Blurring of Pharmacological and Nutritional Frameworks

*Withania somnifera* L. is traditionally consumed as a root extract in dietary supplements to support stress management, sleep, and hormonal balance. However, the DTU assessments failed to distinguish between its pharmacological and nutritional uses, applying pharmaceutical-level toxicological expectations to a botanical widely regarded as safe under traditional and evidence-based guidelines. That is a critical point in DTU assessment, as the study of question [172] involved dry steam of *Withania somnifera* dried 50% ethanol extract, which is a significantly different preparation of roots specified in the Indian and UK Pharmacopeias and used for other indications and health claims.

2.Inappropriate Aggregation of Data from Different Plant Parts

A core flaw in the DTU evaluations is the failure to differentiate between root, leaf, stem, and berry extracts, despite well-established phytochemical and pharmacological differences. This has led to the erroneous attribution of adverse effects (e.g., cytotoxicity from withaferin A, primarily found in leaves) to root-based products, undermining the validity of the conclusions. The spermicidal activity was determined in vitro tests by mixing *Withania somnifera* stem extract with sperm suspension. The mixture was observed under a microscope for 20 s and examined for motile sperm. The minimum effective spermicidal concentrations of *Withania somnifera* stem extract were found to be 10 mg/million sperm in suspension containing 42.5 sperm count/mL. The authors conclude that 50% ethanol-dried *Withania somnifera* stem extract has an antifertility effect on male rat reproduction, sexual behavior, and epididymal sperm concentration. The oral administration of the extract at 50 mg/kg body weight/day in male albino rats produced antifertility effects. The point is that both the authors and DTU experts do not question the bioavailability of this stem extract and the corresponding oral dose in humans. That is another critical point leading to wrong assumptions and decisions to ban Ashwagandha products. For comparison, the concentration of *Withania somnifera* stem extract in that study was at least 1000-fold higher than that of the blood plasma of *Withania somnifera* root extract, as evident from pharmacokinetic studies and the EFSA assessment report [165]. *Withania somnifera* root extract was investigated after a single oral administration of 500 mg/kg in six non-descript healthy buffalo calves. The mean peak plasma concentration at 0.75 h was 248.16 ± 16.12 μg/mL and was detected up to 3 h, with a mean plasma concentration of 6.55 ± 0.12 μg/mL. That is at least 1000-fold lower than the spermicidal activity. The maximal blood plasma concentrations of withaferin A and withanolide A in mice were found to be 16 and 26 ng/mL, respectively, after oral administration of an aqueous extract of ashwagandha root [165].

3.Selective and Outdated Use of Scientific Literature

Although hundreds of clinical and toxicological studies on Ashwagandha have been published in the past decade, many of which demonstrate a favorable safety profile, the DTU assessments relied heavily on older and animal-based data, some of which lacked relevance to human use. Peer-reviewed safety evaluations, including those by the American Herbal Products Association (AHPA), were omitted.

4.Absence of Peer Review and Transparency

Neither of the DTU reports underwent external peer review. Key methodological details—such as criteria for evidence inclusion, levels of uncertainty, and expert qualifications—were not disclosed, raising concerns about the objectivity and reproducibility of the findings.

5.Regulatory Disparity and Industry Impact

Discrepant evaluations of the same botanical substance by different national authorities—some viewing Ashwagandha as a safe food supplement, others banning it outright—have generated regulatory friction and potential trade barriers. This inconsistency particularly disadvantages producers in countries like India, where the plant has been used safely for centuries.

### 2.7. Critical Assessment of Common Technical Documentation Submitted by Drug Manufacturers to Drug Authorities

On the other side, the rejections to grant ‘well-established use’ of some herbal medicines are often based on assessment reports of drug authorities, e.g., EU herbal monograph on *Rhodiola rosea* L., rhizoma et radix [38], which identified serious problems such as:The published clinical trials exhibit considerable deficiencies in their quality and show methodological problems.
○insufficiently characterized herbal preparations,○open (label) studies,○small sample size,○missing ITT analysis, regardless of a detailed description of dropouts and reasons for exclusion in the analysis of an outcome measure,○healthy subjects,○The efficacy score has not been validated○The results from trials on clinical pharmacology are contradictory.○There is a lack of independent replications of the single studies.

The sources of inconsistency of the results of various clinical studies of *Rhodiola rosea* L., rhizoma et radix, are due to the authors not adhering to the consolidated standards of reporting trials (CONSORT) recommendations [125] and ICH E6 (R3) Guideline for good clinical practice (GCP)—Step 5 [126]. For example, in a randomized controlled trial, the authors investigated the effect of *Rhodiola rosea* on mental and physical fatigue in nursing students [165,177]. However, the authors did not characterize the product with respect to extraction solvents and dry herb, dry native extract ratio (DER), the content of active markers, and did not provide HPLC fingerprints to ensure consistent quality and reproducible pharmacological activity. The analytical methods were not validated for selectivity, accuracy, and precision [165,177].

The authors declared that placebo capsules containing microcrystalline cellulose and silicon dioxide had the same appearance, odor, and taste as the R. rosea product, which is very unlikely due to their strong specific rose odor, test, and color, particularly when “participants were asked to self-determine their need for one additional capsule (i.e., a half dose), to be taken within four hours of the initial dose [165,177].

The authors have not reported (or assessed) the results of treatment compliance (counting of unused tablets), which is a serious flaw [165,177].

All outcome measures of the study were subjective based on self-assessment questionnaires of QOL in 48 nurses instead of the only doctor having the same unified “standard” [165,177].

The imbalance between the Rhodiola treatment and placebo groups in medication use and physical and emotional health problems has had a significant impact on the results of the study [165,177].

### 2.8. Other Challenges in Adaptogens Research

It should be noted that some other challenges in adaptogens research are related to the insufficient characterization and standardization of certain active constituents, including the pyrazole alkaloid withasomnine, Appendix A, found in Withania root preparations [43,178,179], which is known for its sedative properties. New studies are required to demonstrate the reproducibility of well-characterized Ashwagandha-containing products.

Another poorly studied subject is the content of cyanogenic terpenoid lotaustraline identified in *Rhodiola rosea* roots [73,180,181], which should be standardized for its content in *Rhodiola rosea* preparations to ensure reproducible and consistent results in various clinical trials [182], Appendix A.

## 3. Pharmaceuticals vs. Nutraceuticals and Dietary Supplements

### 3.1. Regulatory Classification, Purpose of Use, Dose and Potency, Labeling and Claims

The requirements for pharmaceuticals [16,17,18,19,154,157,183,184,185,186,187,188,189,190,191,192,193,194,195,196,197,198,199,200,201,202,203,204,205] and dietary supplements [158,206,207,208,209,210,211,212,213,214,215,216,217,218,219,220,221,222,223,224,225,226,227,228,229,230,231,232,233] of botanical origin differ significantly in terms of regulatory oversight, quality standards, safety and efficacy requirements, manufacturing practices, intended use, health and functional claims, and supported health functions [194,195,196,197,198,199,200,201,202,203,204,205,206,207,208,209,210,211,212,213,214,215,216,217,218,219,220,221,223], see Appendix A and Table 4 and Table 5. Table 6 shows the key differences in the requirements for botanical-origin pharmaceuticals and dietary supplements under FDA (United States) vs. EMA (European Union) regulations. Table 7 includes selected true adaptogens with Pharmacopeial Recognition and some putative adaptogens, such as Eurycoma and Sideritis, which were claimed as adaptogens in publications [107,108,109,110,111,112,113].

The daily dose of a dietary supplement versus a herbal medicine containing the same active ingredient can differ significantly due to regulatory, intended use, and formulation factors. For example, the daily dose of Eleutherococcus dry root and rhizome is 2–3 g, equivalent to a pharmaceutical preparation of quality [18]. The daily dose of Ginseng dry root is 0.6–2 g, equivalent to a pharmaceutical preparation of quality [19].

Table 5 presents the differences in requirements for pharmaceuticals and dietary supplements of botanical origin in Sweden, which follow EU-wide regulations for botanical products but incorporate national layers through two key agencies: the Swedish Medical Products Agency (MPA) for medicinal products and the Swedish Food Agency for dietary supplements [158,204].

Even if the active ingredient is identical, herbal medicines generally have higher, standardized, and therapeutically justified doses. In contrast, dietary supplements offer lower doses for general wellness and are not intended to treat diseases.

In summary, this section shows the differences between the requirements for the quality of pharmaceuticals and dietary supplements of botanical origin.

Pharmaceutical-grade botanicals typically have standardized doses that are higher, aimed at treating and curing diseases, and are subject to strong regulation and evidence-based research. In contrast, dietary supplements typically have lower, less consistent doses, focusing on wellness and safety to support health and nutrition. They are less regulated, as they are not intended to treat disease.

Dietary supplements typically provide lower, variable doses with less standardization, whereas pharmaceutical-grade ginseng products offer standardized, clinically tested doses for specific therapeutic uses.

Rhodiola rosea exhibits differences in dose depending on whether it is marketed as a dietary supplement or a pharmaceutical/herbal medicinal product (Appendix A). Rhodiola as a dietary supplement usually comes in lower, less standardized doses. At the same time, as a pharmaceutical herbal medicine, it is administered at higher, clinically validated, and standardized doses for specific conditions, such as stress-related fatigue. The dosage of ginseng differs significantly depending on whether it is sold as a dietary supplement or as a pharmaceutical-grade herbal medicine. Eleutherococcus products sold as nutritional supplements often have higher variability and may use lower-quality or non-standardized root powder. In contrast, pharmaceutical-quality products are standardized, clinically tested, and typically prescribed at ~300–400 mg/day extract (2–4 g/day root equivalent) for therapeutic purposes. Schisandra supplements typically use lower, non-standardized doses (200–500 mg/day) for wellness purposes, whereas pharmaceutical-quality preparations employ higher, standardized doses (500–1500 mg extract/day or 1.5–6 g dried fruit) to achieve consistent therapeutic effects.

Ashwagandha Dietary supplements usually provide lower, less consistent doses. At the same time, pharmaceutical/medicinal Ashwagandha in India uses higher, standardized, clinically validated doses (often double the typical supplement amount, or even more in traditional powdered form).

In contrast to many herbal medicinal products, such as St. John’s Wort, the doses of other well-established adaptogens, including Ginseng, Rhodiola, Eleutherococcus, and Schisandra preparations, are almost within the same range, regardless of their quality grade: pharmaceuticals or dietary supplements (Table 8).

The reasons for the existing differences came from several issues, including:Insufficient product characterization, lack of product standardization, poor and biased reporting that do not meet CONSORT [125] recommendations, poorly conducted clinical studies that do not meet ICH guidelines for GCP requirements [126], and lack of GMP [197,199].Complex multi-component botanicals, particularly of fixed combinations of herbal preparations with reproducible and sustainable quantitative and qualitative chemical composition. That is a challenge in analytical chemistry and standardization, ensuring the production of acceptable quality products. Appendix A. The more components a product has, the more resources and time are spent on quality control, and the higher the product’s cost.Differences in requirements for manufacturers of medicines and food products in different countries.Some tasks and goals of manufacturers and regulatory authorities are different, which can sometimes lead to ambiguous results or conflicts of interest.

In any case, the most effective way to find the best solution to the problem is a scientific and unbiased approach.

### 3.2. Proposed Solutions

Establish Internationally Harmonized Guidelines for evaluating botanicals based on their intended use (e.g., pharmaceutical vs. dietary Supplement), incorporating traditional use data alongside modern scientific methods.Encourage Peer Review and Transparency in national assessments by mandating public disclosure of methodologies, data sources, and expert affiliations.Create a Tiered Evidence Framework that allows differentiated standards of proof for traditional botanical supplements versus pharmaceutical candidates.Promote International Scientific Dialogs among regulators, researchers, and industry to develop consensus positions and avoid unilateral bans that may lack scientific rigor.

## 4. Discussion

Adaptogens are botanicals that trigger the adaptive stress response of the organism by feedback down-regulating mechanisms, resulting in the normalization of physiological functions and supporting homeostasis. As a result, they have nonspecifically targeted multiple effects in the stress system (neuroendocrine immune complex) associated with the cardiovascular and gastrointestinal systems, which play an essential role in stress-induced mental/ brain fatigue, as well as aging-related diseases.

In essence, adaptogens induce adaptability and exhibit adaptogenic effects via adaptive mechanisms where the key role is cortisol targeting corticosteroid receptors. Other intracellular and extracellular mediators of the stress response, such as ATP and GTP, are involved in regulating the activation and inactivation of canonical signaling pathways in physiological processes within the networks of mediators of the stress response.

Network pharmacology studies have uncovered the mechanisms of action of adaptogens, providing insight into how to maintain homeostasis in various stressful conditions and prevent and cure diseases. Therefore, adaptogens are useful both as pharmaceuticals, dietary supplements, and nutraceuticals. In this context, adaptation, like vitamins, is a distinct category of natural products that should be classified as beneficial agents in the borderline between healthy conditions and diseases, both for normalizing physiological functions and curing diseases. In the background of this classification are experienced traditional medicine systems (Ayurveda, TCM, Kampo) [2], Brekman’s theory of valeology and pharmacosanation [50,51,244,245,246,247], and the elucidation of the mechanism of action of adaptogens through the implementation of network pharmacology analysis/systems biology approaches in adaptogens’ research. The interaction between Western and Oriental medical knowledge results in their integration, leading to beneficial symbioses for new drug development. In this context, harmonizing the food and drug regulatory network is essential for human health.

## 5. Conclusions

Adaptogenic botanicals occupy a complex regulatory space with inconsistent risk assessments. The dual-faced nature of adaptogens necessitates harmonization among Food and Drug regulatory authorities in various countries and the formal recognition of adaptogens as a category of natural products for the prevention of stress-induced brain fatigue, behavioral, and aging-related disorders in individuals with subpar health status. We also conclude that insufficient evidence and differences in the criteria for assessing the quality, efficacy, and safety of botanicals may result in inconsistent results, particularly in clinical studies. The inconsistencies in risk assessments of *Withania somnifera* underscore the broader challenge of regulating botanicals that straddle the food–pharma boundary. Without harmonized standards and transparent methodologies, regulatory decisions risk being unscientific or influenced by conflicts of interest. A more balanced, evidence-informed approach is urgently needed to protect consumer safety without stifling access to beneficial botanicals.

## Figures and Tables

**Figure 1 pharmaceuticals-18-01346-f001:**
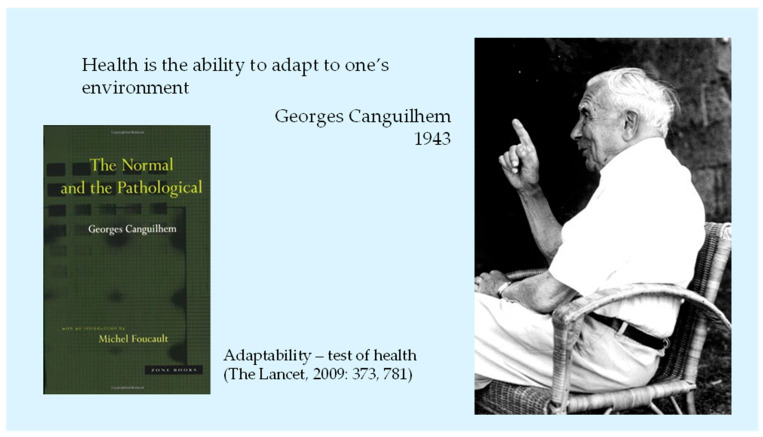
Georges Canguilhem introduced the term adaptability in 1943, who defined adaptability as the ability of an organism to alter itself or its responses to the changed environment or circumstances and assumed that adaptability shows the ability to learn and improve from experience—repeated mild exposure or low doses of stress result in the increased resistance of cells and organisms to subsequent stress exposure, resulting in an adaptation that favors survival [6,7]. According to the WHO definition, “health is the state of complete physical, mental, and social well-being and not merely the absence of disease or infirmity” [58]. Notably, that is formally different from the definition of health distinct by George Canguilhem as “the ability to adapt to one’s environment”, but in line with the WHO.

**Figure 2 pharmaceuticals-18-01346-f002:**
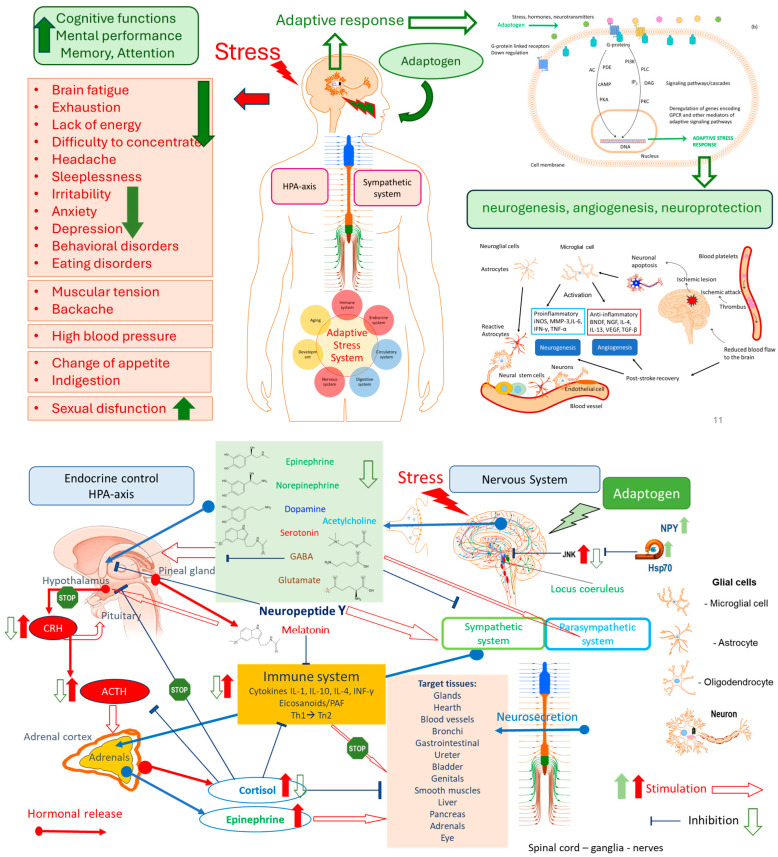
Chronic stress-induced symptoms, as well as the effects of adaptogens on key mediators and effectors of the adaptive stress response, result in neuroprotection, leading to improved cognitive function and enhanced mental and physical performance. Brain cells respond adaptively by enhancing their ability to function and resist stress, as demonstrated by an update from the authors’ free-access publication, along with their accompanying drawings. Simplified overview of the stress system (central nervous system, CNS, and peripheral tissues/organs in the periphery) and reciprocal connections of elements of the neuroendocrine-immune complex to mobilize an adaptive response against the stressor [3].

**Figure 3 pharmaceuticals-18-01346-f003:**
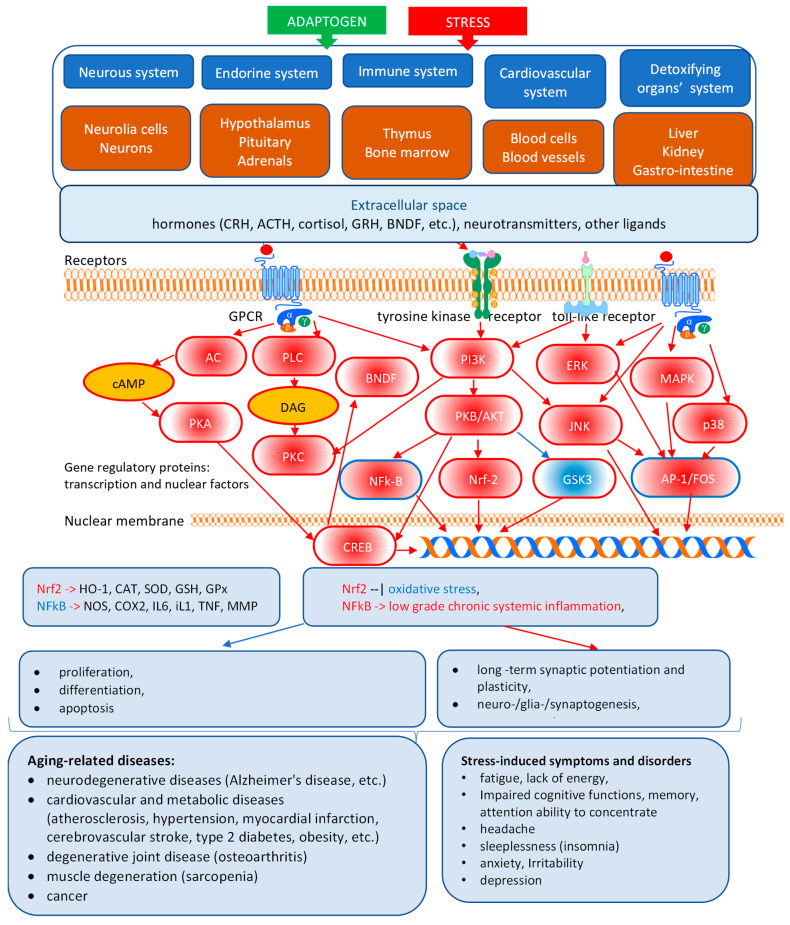
The hypothetical molecular mechanisms and modes of pharmacological action of adaptogens on key mediators of the neuroendocrine-immune complex, cardiovascular, and detoxifying systems that regulate the adaptive stress response to stressors/pathogens in stress- and aging-induced diseases and disorders [3,19]. Reproduced from the authors’ free-access publication [66] and the authors’ drawings. Activation is shown in red, while the inhibition is in blue (effect of ginseng/ginsenosides), cycles/ellipses, arrows, and clouds. BDNF, brain-derived neurotrophic factor; cAMP, cyclic adenosine monophosphate; CREB, cAMP-responsive element-binding protein; ERK, extracellular signal-regulated kinase; GSK-3, glycogen synthase kinase-3; JNK, the c-Jun N-terminal kinase (JNK)/stress-activated protein kinase (SAPK MAPK, mitogen-activated protein kinase; NF-kB, nuclear factor-kappa B; Nrf2, nuclear factor E2-related factor 2; PI3K, phosphatidylinositol 3-kinase; PKA, protein kinase A; PKB, protein kinase B; PLC, phospholipase C.

**Figure 4 pharmaceuticals-18-01346-f004:**
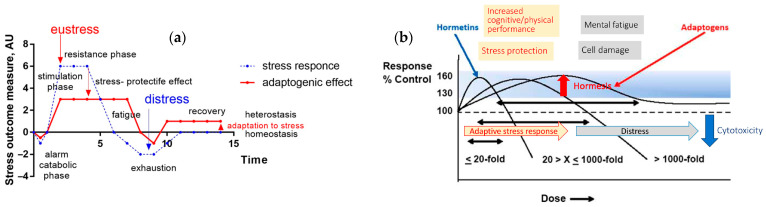
(**a**)—Simplified pattern of adaptive stress response over time (blue color dot line) and stress protective effect of an adaptogen (red color line) increasing resistance and decreasing sensitivity to a conditional stressor, extending the duration of the resistance anabolic phase to fatigue, and preventing the onset of the exhaustion phase (distress), leading to adaptation to stress and increasing the level of homeostasis. AU—an arbitrary unit of an outcome measure of a stress response, e.g., increasing the number of errors over time detected in Conner’s computerized cognitive performance test for attention and impulsivity, biochemical markers (cortisol, nitric oxide, etc.) [79]. It is reproduced and modified from [3,61,66]. (**b**)—Simplified hypothetical description of the hormetic dose–response relationship of toxic hormetins and adaptogens; author’s drawings reproduced from the authors’ free access publication [3].

**Table 3 pharmaceuticals-18-01346-t003:** Basic Principles of Network Pharmacology: Traditional vs. Network Pharmacology.

Feature	Traditional Pharmacology	Network Pharmacology
Philosophy	Reductionist—targets one gene/protein	Systems-oriented—considers multitarget interactions
Target Focus	Single molecule	Multiple targets, often in networks
Drug Design Goal	High specificity	Modulation of networks/pathways
View of Disease	Caused by the dysfunction of a single entity	Disease as a network perturbation
Data Used	Experimental pharmacokinetics/dynamics	Multi-omics, PPI networks, computational modeling
Mechanism Identification	Binding affinity and downstream effects	Topological influence on biological networks
Predictive Capacity	Limited to known targets	Broader scope; includes off-targets, repurposing, synergy predictions
Herbal/TCM Suitability	Not applicable	Especially suitable due to the multi-component nature
Validation	Strong experimental support	Requires computational and experimental integration
Limitations	Ignores complexity and off-target effects	Data noise, oversimplified networks, context-ignorant models

**Table 4 pharmaceuticals-18-01346-t004:** The comparison of differences in requirements for pharmaceuticals and dietary supplements of botanical origin.

Category	Pharmaceuticals [16,17,18,19,154,157,183,184,185,186,187,188,189,190,191,192,193,194,195,196,197,198,199,200,201,202,203,204,205]	Dietary Supplements [158,206,207,208,209,210,211,212,213,214,215,216,217,218,219,220,221,222,223,224,225,226,227,228,229,230,231,232,233]
*Regulatory Oversight*	Very strict	Lenient
Regulated by	Strictly regulated by drug authorities (e.g., FDA *, EMA) **	Governed by food supplement laws (e.g., DSHEA *** in the U.S., EFSA **** in the EU).
Pre-market approval	Required (clinical trials, IND, NDA, etc.)	Not required; must follow labeling and safety guidelines.
*Evidence of Safety and Efficacy*	Required clinical + preclinical	Not required; General safety only
Efficacy	Must be proven through rigorous clinical trials	No requirement to prove efficacy before marketing
Safety	Extensive safety data required (nonclinical + clinical)	Only required to ensure general safety; no clinical trials mandated
*Quality Standards*	High	Moderate
Identity and purity	Must meet strict pharmacopoeial standards (e.g., USP ^†^, EP ^††^)	Less stringent, basic identity and purity testing is often enough, focus on identity, purity, and composition
Standardization	Active ingredients must be quantified and consistent	Often contains a range of components; standardization is not always required
Contaminants (e.g., heavy metals, microbes)	Tightly controlled with established limits	Limits exist, but are less strictly enforced
Batch-to-batch consistency	Mandatory and validated	Expected but not strictly enforced
*Manufacturing Requirements*	Pharmaceutical grade, GMP	Food GMP
GMP Standards	Must follow pharmaceutical GMP (e.g., ICH Q7 ^†††^, EU GMP, Part 211)	Must follow food-grade GMP (e.g., FDA Part 111), which are less stringent
Process validation	Mandatory for all critical manufacturing steps	Not required for all processes
Change control and documentation.	Detailed documentation and validation are required.	Documentation is required, but it is generally simpler.
*Consistency of Botanical Source Specifics*	Standardized	Variable
Botanical identity	Must be rigorously confirmed and controlled	Often confirmed, but methods may vary in rigor
Extraction process	Fully validated and standardized	May vary; often not standardized
Complex mixtures	Defined active constituents or fractions used	Often, a whole plant/extract with variable composition
*Labeling Claims and Doses*	Indications for use in diseases	Health supporting claims
Health claims	Treat/cure diseases; medicine list “Dosage” based on age, weight, or condition. Can make therapeutic claims (e.g., “treats depression”)	Support health functions; Supplements list “Suggested use” or “Serving size”. Cannot make disease claims; only “structure/function” claims (e.g., “supports mood”)
Labeling accuracy	Must match approved documentation	Must be truthful and not misleading, but with less scrutiny
Regulatory Classification	Regulated as a medicine or therapeutic product. It can be used for specific health conditions with evidence to support it. Subject to stricter quality, efficacy, and safety controls. May contain higher or more standardized doses.	Regulated as food in most countries (e.g., by the FDA in the U.S.). Intended for general health support (e.g., to “maintain” or “support” function). Cannot claim to treat, prevent, or cure diseases. Doses are often lower to avoid therapeutic claims or side effects.
Purpose of Use	Medicines are designed for therapeutic effect and are often used for shorter-term or targeted purposes.	Supplements aim to provide nutritional support and are typically used on a long-term basis.

FDA *, EMA **, DSHEA *** in the U.S., EFSA ****, USP ^†^, EP ^††^, ICH Q7 ^†††^.

**Table 5 pharmaceuticals-18-01346-t005:** The comparison of differences in requirements for pharmaceuticals and dietary supplements of botanical origin in Sweden.

Product Type	Herbal Medicinal Products	Botanical Dietary Supplements
Regulator	Swedish Medical Products Agency	Swedish Food Agency,
Key Legislation:	under EU Directive 2001/83/EC and national provisions	○ under EU food law Directive 2002/46/EC on food supplements ○ LIVSFS 2023:3—Swedish Food Agency’s national regulation
Classification:	○ Herbal Medicinal Products ○ Traditional Herbal Medicinal Products	○ Plant extracts ○ Must have physiological effects
Definitions:	○ *Herbal substances*: Whole or parts of plants, algae, fungi, or lichens. ○ *Herbal preparations*: Extracts, powders, distillates, etc.	
Approval Needed	Yes	No (registration only)
Approval Requirements	○ Must be registered or authorized before sale. ○ Traditional herbal products require bibliographic evidence of safety and plausible efficacy over at least 30 years (15 in the EU). ○ Full marketing authorization demands clinical and preclinical data, similar to conventional drugs.	○ No pre-market approval needed, but food business operators must be registered. ○ Products must be safe, properly labeled, and not misleading. ○ Health claims must comply with Regulation (EC) No. 1924/2006.
Claims Allowed	Treat/cure diseases	Support health functions (not a cure)

**Table 6 pharmaceuticals-18-01346-t006:** FDA (United States) vs. EMA (European Union) regulations.

Aspect	FDA (U.S.) [196,197,198,206,207,208,209,210,211,212,213,214,215,216,217,218,219,220,221]	EMA (EU) [157,158,183,184,185,186,187,188,189,190,191,192,193,194,195,222,223,224,225,226,227,228,229,230,231,232,233]
*Governing Bodies*	- Pharmaceuticals: Center for Drug Evaluation and Research (CDER) - Supplements: Center for Food Safety and Applied Nutrition (CFSAN)	- Pharmaceuticals: European Medicines Agency (EMA) - Supplements: Regulated at member state level (e.g., Germany: BfArM, France: ANSM)
*Applicable Legal Frameworks*	- Drugs: FD&C Act, 21 CFR - Supplements: Dietary Supplement Health and Education Act (DSHEA, 1994)	- Drugs: Directive 2001/83/EC - Supplements: Food Supplements Directive (2002/46/EC), national laws
*Botanical Drugs*	Defined as botanical drug products, subject to full NDA or IND path (e.g., Veregen^®^, Mytesi^®^)	Herbal medicinal products (HMPs), classified into:—Well-established use (WEU)—Traditional use (THMP)—Full marketing authorization
*Supplements (Botanical)*	Treated as foods, not drugs. No pre-market approval. No efficacy proof required.	Also treated as foods, but the EU is more restrictive on claims. Heavily influenced by EFSA assessments.
*Quality Standards for Botanicals*	Encourages use of USP monographs and FDA Botanical Drug Guidance (2004). Must define active constituents or marker compounds.	Uses European Pharmacopeia (Ph. Eur.) monographs. Strict on identity, purity, and standardization. The Herbal Medicinal Products Committee (HMPC) oversees scientific guidelines.
*Clinical Evidence (Botanical Drugs)*	IND → NDA process: requires full clinical trials unless eligible for accelerated approval.	WEU: requires published literature and some clinical data. THMP: based on 30 years of traditional use (15 in the EU), with nonclinical safety evidence only
*Labeling (Supplements)*	Structure/function claims allowed: “supports immune health.” Must carry a disclaimer: “This product is not intended to diagnose, treat, cure, or prevent any disease.”	Health claims reviewed and authorized by EFSA; therapeutic claims prohibited on supplements. Stricter than the FDA.
*GMP*	- Drugs: 21 CFR Part 210/211- Supplements: 21 CFR Part 111	- Drugs: EU GMP (Annexes)- Supplements: Food GMP (varies by country); less
*Unique Points*	Allows botanical drug development via standard drug approval paths. Dietary supplements are widely available with relatively light regulation, provided safety is ensured. Botanical Drug Development Guidance has been available since 2004.	Provides a specific regulatory framework for traditional herbal medicinal products (THMPs) via simplified registration. More centralized regulation of herbal drugs via EMA’s HMPC. Supplements are subject to tighter control over labeling and claims, often stricter than in the U.S.
*Example*		
Echinacea supplement	Dietary Supplement, no pre-market approval	Food supplement; cannot claim therapeutic effects
Echinacea extract as a medicine	Must go through the full IND/NDA process	Can qualify as THMP or WEU based on evidence and monograph

**Table 7 pharmaceuticals-18-01346-t007:** Countries with Pharmacopeial recognition of selected adaptogens [234,235,236,237,238,239,240] and some putative adaptogens, such as Eurycoma and Sideritis [241,242,243], and their official regulatory status: ✅—Official monographs; 
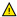
—Dietary supplement use.

Country	PharmacopeiaMonographs	Rhodiola	Ginseng	Withania	Eleuthero-Coccus	Schisandra	Eurycoma	Sideritis
Russia	State Pharmacopeia [189,190]	✅	✅		✅	✅		
China *	Pharmacopeia of PRC -2010 (Eng) [191]		✅		✅	✅		
European Union	EP8 European EMA/HMPC Union herbal monograph	✅	✅	✅	✅	✅		✅
United States	USP, USP Herbal Compendium monograph, AHP	✅	✅	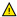	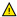	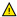		
Germany	Commission E	✅	✅	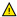	✅	✅		
UK	British Pharmacopoea		✅	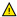	✅			
Mongolia	Mongolian Pharmacopeia	✅			✅	✅		
India *	Indian Herbal Pharmacopeia	✅		✅				
Pakistan	Unani/Ayurvedic Pharmacopeia			✅				
Bangladesh	Unani Pharmacopeia			✅				
Sri Lanka	Ayurvedic Pharmacopeia			✅				
South Korea	Korean Herbal Pharmacopeia		✅		✅	✅		
Japan	Japanese Pharmacopeia	✅	✅		✅ 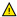	✅		
Vietnam	Vietnamese Pharmacopeia		✅				✅	
Lanka								
Australia	Australian Register of Therapeutic Goods (ARTG)			✅				
Malaysia	Malaysian Pharmacopeia						✅	
Indonesia	Indonesia Pharmacopeia						✅	
South Africa	CAM regulatory framework			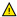				
WHO	WHO Monographs		✅	✅	✅			

*—Appendix A.

**Table 8 pharmaceuticals-18-01346-t008:** The comparison of doses for adaptogenic pharmaceuticals and dietary supplements.

Botanical	Dietary Supplement Dose	Pharmaceutical Dose
Ginseng	100–400 mg/day extract (variable, not always standardized)	200–400 mg/day standardized extract (1–2 g/day root equivalent)
Rhodiola	100–300 mg/day extract (variable)	200–600 mg/day standardized extract (3% rosavins, ~1% salidroside)
Eleutherococcus	300–1200 mg/day crude root or extract (variable)	300–400 mg/day standardized extract (2–4 g/day root equivalent)
Schisandra	200–500 mg/day extract (variable)	500–1500 mg/day standardized extract OR 1.5–6 g/day dried fruit
Ashwagandha	300–600 mg/day extract (often not standardized)	500–1000 mg/day standardized extract (3–6 g/day root powder in Ayurveda)
Rhodiola	100–300 mg/day extract (variable)	200–600 mg/day standardized extract (3% rosavins, ~1% salidroside)
St. John’s Wort	100–300 mg/day extract (variable)	900 mg/day standardized extract (hypericin/hyperforin defined)

## Data Availability

Not applicable.

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
