# Peer review of "Two Sides of the Same Coin for Health: Adaptogenic Botanicals as Nutraceuticals for Nutrition and Pharmaceuticals in Medicine"

_pharmaceuticals, 2025, doi:10.3390/ph18091346_

Round 1
Reviewer 1 Report
Comments and Suggestions for Authors
- Please clarify the definition or context of health as used in manuscript.
- Does the manuscript offer any recommendations regarding optimal patient selection, dosage regimen, or specific formulations of adaptogens for further research needed?
- The structure of the manuscript is not clear. Sections are not well-defined and there is a lack of logical flow and continuity between various sections and subsections.
- Many changes are required, mostly for the provision of sources (references) for many claims (Table 1, 2, and 3).
- How is the current level of evidence graded for somnifera therapy using accepted frameworks like GRADE or Oxford CEBM Levels of Evidence?
- What evidence syntheses or meta-analyses have already been published combining data from multiple clinical trials of herbal medicines?
- The authors did not highlight the role of secondary metabolites of plants in the treatment of disease.
- Which specific chemical or analytical tests are employed to detect adaptogens prior to their use as therapeutic agent?
- Identify and elaborate on the particular classes of phytochemicals present in somnifera that contributes to its different pharmacological properties.
- Describe the figure 3 and 4 in the main text to enhance compehension.
- A brief summary should be provided at the end of each major section.
- Please clarify the rationale or evidence behind the claim that botanical adaptogens are more efficient in treating human diseases than conventional allopathic medicines?
- Include any available data or findings from clinical trials involving adaptogens?
- Specify the type of review?
- Abstract: Add future perspectives in the conclusion.
- Use of botanical substances does not look good. I suggest for using plant-derived natural products.
- Please check lines 41-43.
Comments on the Quality of English Language
Some sentences are grammatically inapropriate.
Reviewer 2 Report
Comments and Suggestions for Authors
I have read, reviewed and evaluated the MS titled “Two Sides of the Same Coin for Health: Adaptogenic Botanicals as Nutraceuticals for Nutrition and Pharmaceuticals in Medicine” for possible publication in Pharmaceuticals. Authors should strictly follow all my suggestions/comments to enhance the quality of the manuscript.
Abstract
- The abstract outlines the scope and conclusions clearly, but it is somewhat general. It could be improved by explicitly defining adaptogens (e.g. as herbs that “enhance the state of non-specific resistance” to orient readers immediately to the topic. It currently speaks of “dual-faced” botanicals and proposed solutions without specifying the key concepts.
- The aims and solutions (harmonized guidelines, transparency, tiered evidence, etc.) are reasonable. However, they are stated as given rather than demonstrated. It would strengthen the abstract to indicate on what basis these solutions are proposed. For example, citing a regulatory source or prior study that identifies these problems would add authority.
- The conclusions (“harmonized standards, transparent methodologies…balanced, evidence-informed”) flow from the abstract but are broad. The abstract might note that this is a narrative overview rather than new data, and possibly mention any major limitations (e.g. reliance on existing literature, which may be uneven).
Introduction
- The introduction sets the stage by noting regulatory ambiguity of botanicals, but it repeats the phrase “dual-faced nature” twice in nearly identical ways (lines 82–90 which seems redundant. Tightening this section to avoid repetition would improve clarity.
- Importantly, the introduction fails to define “adaptogen” before using it. To be precise and balanced, the authors should include a concise definition early on (for example, as “a substance that increases non-specific resistance to stress” and note that this concept, while rooted in traditional and Soviet medicine, is not formally recognized by most modern regulators.
- The scope is generally clear, but the narrative jumps to conclusions about “conflicts of interest between industry and regulators” without evidence. It would help to frame these as concerns or hypotheses unless a source is cited. For example, if industry lobbying is alleged to influence outcomes, a reference (or at least an example) should be provided. Otherwise, such statements read as opinion.
- Pharmaceuticals vs. Nutraceuticals and Dietary Supplements
- This section is well organized with helpful tables. Table 1 (pharmaceutical vs supplement quality requirements) and Table 2 (FDA vs EMA) capture key differences clearly. However, many claims in the text and tables are presented without citation. For example, the statement that FDA’s botanical drug guidance encourages using USP monographs and defining active constituents is accurate, but should be backed by a source. The FDA’s 2004 Botanical Drug Guidance indeed emphasizes defining active markers and controlling raw materials, which could be cited to support the claims about strict pharmaceutical standards.
- The US/EU focus is appropriate given the journal’s audience, but the authors may consider acknowledging other jurisdictions or clarifying that this comparison is US/EU-centric. For instance, they mention ASEAN regulatory work – it might be useful to cite such examples (e.g. Ramadoss 2022) where relevant or state that they focus on FDA/EMA for brevity.
- In Table 2’s text block, the summary of FDA vs EMA regulations is detailed and mostly accurate. A few points could use clarity or sourcing: e.g., the FDA requires structure/function claims to carry the “diagnose/treat” disclaimer (True, under 21 CFR 101.93), and EFSA indeed authorizes specific health claims, making EU claim rules stricter. Citing the Dietary Supplement Health and Education Act or EFSA claims register could strengthen these statements.
- Table 3 (Pharmacopeial monographs of adaptogens) is interesting but dense. It lists countries and adaptogens with checkmarks. It would be helpful to cite sources for this information (e.g. WHO monographs or national pharmacopeias). For example, authors note Ashwagandha in India and WHO pharmacopoeia. The Patwardhan 2024 article confirms that “Ashwagandha…is recognized in several pharmacopoeias worldwide, such as the Ayurvedic, Indian, British, European, USP, Chinese Pharmacopoeia, and WHO monographs, citing this (or WHO monographs) would validate entries like the checkmark for Ashwagandha in those regions.
- The narrative statement “herbal medicines generally have higher, standardized, and therapeutically justified doses” is true but again should be supported. One might cite a pharmacology or pharmacopeial source to contrast dosing (e.g. the Indian Herbal Pharmacopeia specifying 2–6 g root powder). In its absence, this reads as common knowledge, which is acceptable but less authoritative.
- Progress, Trends, Pitfalls, and Challenges in Adaptogen Research
3.1–3.2 (Adaptogens as Stress Protectors and Stimulants)
- The authors describe adaptogens as acting primarily via mild hormesis: repeated doses trigger adaptive stress-response pathways, whereas a single dose acts as a mild stimulant. These ideas align with the classical definition of adaptogens (increasing the “resistance phase” of stress). It would strengthen the review to cite a source for this concept. For example, Panossian & Wikman (2010) discuss that adaptogens modulate the HPA axis and stress mediators, which supports the authors’ claim here.
- The statement “adaptogens are mild stress-agonists and not stress-antagonists” is thought-provoking, but it might confuse readers without context. If the authors use this pharmacological analogy, they should justify it or cite a reference explaining it. Right now it is presented as fact. Similarly, the use of “eustressor” (positive stressor) is good, but the text should clarify that this theory comes from dose-response research on herbs like Eleutherococcus (cf. Brekhman’s work).
- The claim that a single dose enhances alertness and performance (noted in sports medicine) is plausible (some ginseng and Rhodiola studies show acute effects), but again needs evidence. If possible, the authors should cite at least one clinical trial or review demonstrating short-term performance enhancement. Otherwise, this point stands as an assertion.
- The text in 3.2 is somewhat dense. Phrases like “the stress-protective effect…is not the result of inhibiting the stress response but of adaptive changes” could be clearer. I suggest rewording for readability, e.g., “Instead of simply blocking stress, adaptogens are thought to induce small stress-like stimuli that over time lead to increased resilience. In pharmacological terms, they function as mild stress mimetics, triggering adaptive gene expression in the HPA axis.”
3.3 (Criteria for an Adaptogen)
- The section correctly notes that classical criteria for an adaptogen include non-toxicity, nonspecific enhancement of resistance, and normalizing effect. However, the manuscript only implicitly mentions these (attributed to a definition “in the Amendment below”). It would help the reader if the authors explicitly listed the criteria somewhere in this section (perhaps citing WHO 1964 criteria, or Brekhman & Dardimov’s postulates). They hint at it with “increase resistance to physical and emotional stress”, which is on point.
- The discussion of chronic unpredictable stress (CUS) models and neural changes (line 553–560) is insightful, but possibly too detailed. It explains how stress affects neurons in mice, which establishes relevance of stress models. It might be sufficient to summarize that CUS models induce fatigue/anxiety-like phenotypes, and adaptogens are tested in these models. The electrophysiology detail (D1/D2-PYR cells) is interesting but perhaps beyond the scope of this narrative. Shortening it or moving it to supplementary could improve focus.
- The text then jumps into a list of intracellular pathways (GPCR, TRK, Toll-like, PI3K) activated by adaptogens. This is valuable mechanistic insight, but again quite dense. Since the audience is broad, the authors might summarize: “Adaptogens are reported to activate multiple cell-survival pathways (e.g. via Nrf2, NF-κB, HSP70, and various neuroendocrine mediators), which helps protect neurons under stress.
- Overall, section 3.3 could be reorganized for clarity: begin by stating essential adaptogen criteria (with a citation), then briefly mention animal models used for proof-of-concept, rather than mixing anatomy and molecular details in the main text.
3.4 (Progress and Trends in Adaptogens Research)
- This subsection (partially visible) seems to cover advances like network pharmacology, metabolomics, transcriptomics, etc. It is good to highlight that modern “omics” methods are being applied. However, it would benefit from a balanced tone. For instance, the authors should acknowledge that network analysis, while promising, is still largely theoretical; they might add that “in silico network models can generate hypotheses but require experimental validation.
- If they claim significant progress (e.g. elucidation of pathways), they should provide examples or references. Conversely, if they find no solid results, that too should be stated. The text as given just introduces the concept; a few sentences assessing how useful these trends have been would make it more insightful.
- Place the below text in Section 3.4: Progress and Trends in Adaptogens Research, where the manuscript discusses the role of metabolomics, extraction techniques, and standardization in influencing biological outcomes. Please cite the provided DOI from where the text was adapted. It would smoothly follow or precede the paragraph beginning with………..Unlike conventional medicines, botanical adaptogens consist of multi-component active compounds and phytochemicals, whose interactions lead to novel and unexpected pharmacological activities……..“Emerging research also highlights how extraction methods can significantly influence the physicochemical and functional characteristics of bioactive plant-derived components, such as dietary fibers from Rubus chingii Hu., suggesting that extraction processes may similarly impact the efficacy and bioavailability of botanical adaptogens (doi: https://doi.org/10.1016/j.jff.2022.105081).
3.5 (Pitfalls: Misidentified Adaptogens, Eurycoma Example)
- The authors use Eurycoma longifolia (Tongkat Ali) as a case study of a plant often touted as an adaptogen but lacking evidence. This detailed critique is one of the paper’s strengths: it systematically shows that claims of adaptogenic activity are not supported by rigorous data. For example, they correctly note that traditional uses (aphrodisiac, antidiabetic, etc.) do not by themselves qualify it as an adaptogen.
- The discussion of specific studies (e.g., Talbott 2013) is instructive. The authors point out that important information (standardization of extracts, assay validation) is missing from many publications. This highlights poor trial reporting. To improve this part, the authors could abstract the key issues into a concise table or list. They already do list numerous flaws (e.g., lack of randomization, no ITT, no blinding, etc. in lines 757–776) – these could be formatted as a compact table of “Common Methodological Flaws” for readability.
- The tone here is somewhat harsh (e.g., “there is nothing in that study about … purity”). While accuracy is important, reviewers may prefer more neutral phrasing (e.g., “Critically, the Talbott trial did not report analytical validation of its markers, which raises concerns about extract standardization”). This is more a style point.
- Finally, while focusing on Eurycoma, it might be fair to acknowledge if any solid evidence exists (for instance, the 2017 systematic review [32]). Are there any moderate-quality trials to balance the critique? If not, the authors’ conclusion (“no convincing evidence for adaptogen status”) is justified.
- Citations: This section mostly critiques others’ trials. If possible, referencing guideline documents like CONSORT (for quality standards) or ICH-Q7 (for GMP, noted in Table 1) would bolster their standards. For example, pointing out that the trial did not follow ICH or CONSORT would make the critique more universal. The FDA guidance [25] could again be relevant here about standardization needs.
3.6 (Withania somnifera Ashwagandha Case Study)
- This case study thoroughly summarizes Ashwagandha’s regulatory status in many countries (India, China, USA, UK, Australia, France, Germany, Poland, Sweden, Denmark, ASEAN). It’s impressively comprehensive. The information matches known facts: e.g., Ashwagandha is indeed in the Indian Herbal Pharmacopeia and WHO monographs, and German BfR did express caution.
- Each bullet point listing a country’s status should cite a source. For instance, the Indian Herbal Pharmacopeia inclusion is standard knowledge but could cite the actual pharmacopeia or WHO monograph. The note about 320 ARTG entries in Australia should reference the TGA. The claim about Poland limiting withanolides to 10 mg likely comes from an EU risk assessment and should be cited (EFSA or a national regulation). Providing at least one citation per bullet would greatly strengthen these statements.
- The manuscript then discusses the Danish DTU risk assessments (2020/2023) that led to national bans. The authors take a clear stance that these assessments were flawed. This is a valid point of debate, but it should be presented carefully. The references they give ([1–5,52–62]) include Patwardhan et al. 2024, which sharply criticizes DTU (calling it “far from a critical review”). To maintain neutrality, the authors could summarize DTU’s findings (“DTU claimed possible thyroid and reproductive effects at high doses”) then note the criticisms. Right now, the tone is somewhat dismissive of DTU without fully explaining why DVFA asked for the report. It might help to acknowledge that “the Danish agency cited concerns about hormonal effects and lack of a well-defined safe dose,” and then say that this focus was controversial.
- If possible, add a reference to any official statement by Denmark or EFSA. The McGill blog mentions that Denmark said “impossible to find a safe dose”. Citing [42] or [43] for that summary would contextualize. For example: “As noted by Denmark’s DVFA, its report argued no safe dose could be established with existing data, a conclusion some experts (including Patwardhan et al. 2024) find unconvincing.”
- The point is made that EMA and FDA have not banned Ashwagandha (correct), but the authors might cite that explicitly. For example, they could say “neither the EMA nor FDA have taken similar actions (Ashwagandha remains a permitted supplement in both regions).” The McGill article notes as of mid-2023 that the FDA/EMA did not follow Denmark (can cite [42†L143-L151] if needed, or a statement from FDA/EMA sites if available).
3.7 Key Issues Identified and 3.8 Documentation Assessment
- The listed “Key Issues” (blurring pharmaceut/food frameworks, mixing plant parts, outdated literature use, lack of peer review, regulatory disparity) neatly summarize the earlier discussion. These are well phrased. A minor suggestion is to ensure each bullet stays brief; for instance, the text for bullet 2 runs off the page due to table formatting (lines 1026-1033). Breaking long bullets or reflowing text would improve readability.
- In Section 3.8, the authors note that regulators often reject herbal WEU applications due to poor study quality. They list specific deficiencies (non-standardized extracts, lack of blinding, etc.). This is a useful counterpoint to their criticisms of DTU. To strengthen it, the authors should cite the actual regulatory assessment (e.g. HMPC Assessment Report on Rhodiola 2023 or Panax) from which these criticisms come. They cite [67] (HMPC Rhodiola) but do not list it in the text snippet – ensure consistency with the reference list. Also, a general reference to CONSORT or ICH-Q7 could be included (e.g. “These issues violate CONSORT standards for RCTs” or FDA’s botanical CMC guidelines).
- The discussion is somewhat hard to follow because it mixes two topics (regulator critique vs manufacturer critique). It may help to split 3.7/3.8 with clearer subheadings or lead sentences. For example, start 3.7 with “On the regulatory side, we identify the following recurring issues…” and 3.8 with “On the manufacturers’ side, common deficiencies include…”.
3.9 Other Challenges & 3.10 Proposed Solutions
- The mention of specific constituents like withasomnine and lotaustralin (lines 1132–1143) is a valid point: variability in minor compounds can affect outcomes. This is a strength. It may be worth briefly explaining the significance (“withasomnine is sedative, lotaustralin is a cyanogen; their variability could alter the herb’s effects and safety”).
- The Proposed Solutions bullet list (international guidelines, transparency, tiered evidence, dialogue) is concise and actionable. It reflects common calls in herbal regulatory debates. For completeness, the authors might reference any existing frameworks that could be models (e.g. WHO’s guidelines on herbal medicines, ICH guidelines, the US Office of Dietary Supplements initiatives, etc.). This would show their proposals build on known standards.
- A minor note: ensure that all sub-bullets are consistent in style and tense. For example, solution (i) is “establish…”, (ii) “encourage…”, (iii) “create…”, (iv) “promote…”. This is fine, but (i) is split across lines; consider reformatting for parallel structure.
Conclusions
- The conclusions (as stated in the abstract) emphasize harmonization and evidence-based approaches. Since the manuscript has no separate conclusion section, it might be helpful to add one. A concise concluding paragraph in the main text could restate the main findings (e.g. “We found that adaptogenic botanicals occupy a complex regulatory space with inconsistent risk assessments”) and re-affirm key recommendations.
- Currently, the “Conclusions” are only in the abstract. For the reader’s benefit, a short final section would reinforce take-home messages.
References
- The reference list is extensive (70+ entries) and includes many relevant sources. However, it is heavily weighted toward the authors’ own publications and their collaborators’ work. This in itself is not wrong, but the review would benefit from including more diverse perspectives. In particular:
- Citing official FDA/EMA guidance or scientific opinions would substantiate many claims. For example, the authors might cite the actual FDA Botanical Drug Guidance (2004) or specific EMA/HMPC assessment reports (some are listed, but not all statements have a reference).
- There are few systematic reviews of adaptogen efficacy listed. For balance, citing an independent review (if available) or at least an EFSA evaluation of related claims would be instructive. For instance, EFSA’s stance on ginseng or Rhodiola health claims (Article 13) could be mentioned.
- The tables discuss pharmacopeial recognition. Where a monograph is said to exist (e.g. Indian Herbal Pharmacopeia, USP herb compendium), a reference should be given (or at least “USP 2024 monograph on Ashwagandha” etc.).
- The authors cite a doctoral thesis [32] and some conference abstracts. It would be better, if possible, to replace these with peer-reviewed sources.
Reviewer 3 Report
Comments and Suggestions for Authors
Plant matrices are sources of essential nutrients and bioactive compounds, such as fiber and antioxidants. Therefore, they offer numerous benefits in various sectors, such as food, cosmetics, and pharmaceuticals. However, the regulation of their use presents contradictions and variations from country to country. This review provides an overview of various aspects: sources of inconsistencies in the evaluation of evidence, ensuring the safety, efficacy, and quality of botanical products, and proposing potential solutions to address these issues, particularly in the field of multicomponent botanical products, such as adaptogens—substances, primarily of plant origin, believed to help the body manage stress and maintain balance, promoting resistance to both physical and mental stressors.
The review addresses interesting and highly topical topics, but also difficult to manage, both due to the various aspects covered and the lack of universal guidelines.
Overall, the review provides satisfactory information, and the bibliographical references are adequate.
However, authors should address some issues and improve the flow of their manuscript, as noted below.
The Introduction section shows a probable interruption (line 42): Please check.
Tables 1, 2, and 3 should be introduced with more information.
Line 387: The name of the plant, Withania somnifera, should be italicized: Please check the entire text.
Finally, authors are encouraged to proofread the entire manuscript to eliminate repeated sentences and any inconsistencies.
Round 2
Reviewer 1 Report
Comments and Suggestions for Authors
I would like to thank the authors for addressing my initial comments. The authors have very effectively addressed all comments. Following the revision to the article, I feel that this manuscript is now acceptable for publication.
Reviewer 2 Report
Comments and Suggestions for Authors
I kindly request that you carefully follow all genuine reviewer comments and suggestions without cherry-picking. Please understand that reviewers invest significant time and effort to provide constructive feedback with the sole aim of improving the quality of your work.
In particular, I ask that you address the following comment raised in my previous review:
Please place the below text in Section 3.4: Progress and Trends in Adaptogens Research, where the manuscript discusses the role of metabolomics, extraction techniques, and standardization in influencing biological outcomes. Cite the provided DOI from which the text was adapted.
This paragraph should smoothly follow or precede the one beginning with:
“Unlike conventional medicines, botanical adaptogens consist of multi-component active compounds and phytochemicals, whose interactions lead to novel and unexpected pharmacological activities…”
Text to insert:
“Emerging research also highlights how extraction methods can significantly influence the physicochemical and functional characteristics of bioactive plant-derived components, such as dietary fibers from Rubus chingii Hu., suggesting that extraction processes may similarly impact the efficacy and bioavailability of botanical adaptogens (doi:10.1016/j.jff.2022.105081).”
Round 3
Reviewer 2 Report
Comments and Suggestions for Authors
I Accept the revised MS for publication